# CONTEXT-AWARE SPARSE COORDINATION GRAPHS

**Tonghan Wang**[1,*]**, Liang Zeng**[1,*]**, Weijun Dong**[1]**, Qianlan Yang**[1]**, Yang Yu**[2]**, Chongjie Zhang**[1]

[1]Institute for Interdisciplinary Information Sciences (IIIS), Tsinghua University
[2]National Key Laboratory of Novel Software Technology, Nanjing University
`tonghanwang1996@gmail.com`
`{zengl18, dwj18, ygl18}@mails.tsinghua.edu.cn`
`yuy@nju.edu.cn, chongjie@tsinghua.edu.cn`

## ABSTRACT

Learning sparse coordination graphs adaptive to the coordination dynamics among agents is a long-standing problem in cooperative multi-agent learning. This paper studies this problem and proposes a novel method using the variance of payoff functions to construct context-aware sparse coordination topologies. We theoretically consolidate our method by proving that the smaller the variance of payoff functions is, the less likely action selection will change after removing the corresponding edge. Moreover, we propose to learn action representations to effectively reduce the influence of payoff functions' estimation errors on graph construction. To empirically evaluate our method, we present the Multi-Agent COordination (MACO) benchmark by collecting classic coordination problems in the literature, increasing their difficulty, and classifying them into different types. We carry out a case study and experiments on the MACO and StarCraft II micromanagement benchmark to demonstrate the dynamics of sparse graph learning, the influence of graph sparseness, and the learning performance of our method[1].

## 1 INTRODUCTION

Many real-world problems involve the cooperation of multiple agents, such as unmanned aerial vehicles (Pham et al., 2018; Xu et al., 2018) and sensor networks (Stranders et al., 2009). Like in single-agent settings, learning control policies for multi-agent teams largely relies on the estimation of action-value functions, no matter in value-based (Sunehag et al., 2018; Rashid et al., 2018; 2020) or policy-based approaches (Lowe et al., 2017; Foerster et al., 2018; Wang et al., 2021c). However, learning action-value functions for complex multi-agent tasks remains a major challenge. Learning individual action-value functions (Tan, 1993) is scalable but suffers from learning non-stationarity because it treats other learning agents as part of its environment. Joint action-value learning (Claus & Boutilier, 1998) is free from learning non-stationarity but requires access to global information that is often unavailable during execution due to partial observability and communication constraints.

Factored Q-learning (Guestrin et al., 2002a) combines the advantages of these two methods. Learning the global action-value function as a combination of local utilities, factored Q functions maintain learning scalability while avoiding non-stationarity. Enjoying these advantages, fully decomposed Q functions significantly contribute to the recent progress of multi-agent reinforcement learning (Samvelyan et al., 2019; Wang et al., 2021b). However, when fully decomposed, local utility functions only depend on local observations and actions, which may lead to miscoordination problems in partially observable environments with stochastic transition functions (Wang et al., 2020; 2021a) and a game-theoretical pathology called relative overgeneralization (Panait et al., 2006; Böhmer et al., 2020). Relative overgeneralization renders optimal decentralized policies unlearnable when the employed value function does not have enough representational capacity to distinguish other agents' effects on local utility functions.

Coordination graphs (Guestrin et al., 2002b) provide a promising approach to solving these problems. Using vertices to represent agents and (hyper-) edges to represent payoff functions defined over the

---

*Equal Contribution.
[1]The MACO benchmark and codes are publicly available at `https://github.com/TonghanWang/CASEC-MACO-benchmark`.

joint action-observation space of the connected agents, a coordination graph expresses a higher-order value decomposition among agents. Finding actions with the maximum value in a coordination graph can be achieved by distributed constraint optimization (DCOP) algorithms (Cheng, 2012), which consists of multiple rounds of message passing along the edges. Recently, DCG (Böhmer et al., 2020) scales coordination graphs to large state-action spaces, shows its ability to solve the problem of relative overgeneralization, and obtains competitive results on StarCraft II micromanagement tasks. However, DCG focuses on predefined static and dense topologies, which largely lack flexibility for dynamic environments and induce intensive and inefficient message passing.

The question is how to learn dynamic and sparse coordination graphs sufficient for coordinated action selection. This is a long-standing problem in multi-agent learning. Sparse cooperative Q-learning (Kok & Vlassis, 2006) learns value functions for sparse coordination graphs, but the graph topology is static and predefined by prior knowledge. Zhang & Lesser (2013) propose to learn minimized dynamic coordination sets for each agent, but the computational complexity grows exponentially with the neighborhood size of an agent. Recently, Castellini et al. (2019) study the representational capability of several sparse graphs but focus on random topologies and stateless games. In this paper, we push these previous works further by proposing a novel deep method that learns context-aware sparse coordination graphs adaptive to the dynamic coordination requirements.

For learning sparse coordination function graphs, we propose to use the variance of pairwise payoff functions as an indicator to select edges. Sparse graphs are used when selecting greedy joint actions for execution and the update of Q-function. We provide a theoretical insight into our method by proving that the probability of greedy action selection changing after an edge is removed decreases with the variance of the corresponding payoff function. Despite the advantages of sparse topologies, they raise the concern of learning instability. To solve this problem, we further equip our method with network structures based on action representations for utility and payoff learning to reduce the influence of estimation errors on sparse topologies learning. We call the overall learning framework Context-Aware SparsE Coordination graphs (CASEC).

For evaluation, we present the Multi-Agent COordination (MACO) benchmark. This benchmark collects classic coordination problems raised in the literature of multi-agent learning, increases their difficulty, and classifies them into 6 classes. Each task in the benchmark represents a type of problem. We carry out a case study on the MACO benchmark to show that CASEC can discover the coordination dependence among agents under different situations and to analyze how the graph sparsity influences action coordination. We further show that CASEC can largely reduce the communication cost (typically by 50%) and perform significantly better than dense, static graphs and several alternative methods for building sparse graphs. We then test CASEC on the StarCraft II micromanagement benchmark (Samvelyan et al., 2019) to demonstrate its scalability and effectiveness.

## 2 BACKGROUND

In this paper, we focus on fully cooperative multi-agent tasks that can be modelled as a **Dec-POMDP** (Oliehoek et al., 2016) consisting of a tuple $G=\langle I, S, A, P, R, \Omega, O, n, \gamma \rangle$, where $I$ is the finite set of $n$ agents, $\gamma \in [0, 1)$ is the discount factor, and $s \in S$ is the true state of the environment. At each timestep, each agent $i$ receives an observation $o_i \in \Omega$ drawn according to the observation function $O(s, i)$ and selects an action $a_i \in A$. Individual actions form a joint action $\boldsymbol{a} \in A^n$, which leads to a next state $s'$ according to the transition function $P(s'|s, \boldsymbol{a})$, a reward $r = R(s, \boldsymbol{a})$ shared by all agents. Each agent has local action-observation history $\tau_i \in \mathrm{T} \equiv (\Omega \times A)^* \times \Omega$. Agents learn to collectively maximize the global return $Q_{tot}(s, \boldsymbol{a}) = \mathbb{E}_{s_{0:\infty}, a_{0:\infty}}[\sum_{t=0}^{\infty} \gamma^t R(s_t, \boldsymbol{a}_t)|s_0 = s, \boldsymbol{a}_0 = \boldsymbol{a}]$.

In a **coordination graph** (Guestrin et al., 2002b) $\mathcal{G} = \langle \mathcal{V}, \mathcal{E} \rangle$, each vertex $v_i \in \mathcal{V}$ represents an agent $i$, and (hyper-) edges in $\mathcal{E}$ represent coordination dependencies among agents. In this paper, we consider pairwise edges, and such a coordination graph induces a factorization of the global Q:

$$Q_{tot}(\boldsymbol{\tau}, \boldsymbol{a}) = \frac{1}{|\mathcal{V}|} \sum_i q_i(\tau_i, a_i) + \frac{1}{|\mathcal{E}|} \sum_{\{i,j\} \in \mathcal{E}} q_{ij}(\boldsymbol{\tau}_{ij}, \boldsymbol{a}_{ij}), \tag{1}$$

where $q_i$ and $q_{ij}$ is *utility* functions for individual agents and pairwise *payoff* functions, respectively. $\boldsymbol{\tau}_{ij} = \langle \tau_i, \tau_j \rangle$ and $\boldsymbol{a}_{ij} = \langle a_i, a_j \rangle$ is the joint action-observation history and action of agent $i$ and $j$.

Within a coordination graph, the greedy action selection required by Q-learning can not be completed by simply computing the maximum of individual utility and payoff functions. Instead, distributed constraint optimization (DCOP) (Cheng, 2012) techniques can be used. **Max-Sum** (Stranders et al., 2009) is a popular implementation of DCOP, which finds optimal actions on a coordination graph $\mathcal{G} = \langle \mathcal{V}, \mathcal{E} \rangle$ via multi-round message passing on a bipartite graph $\mathcal{G}_m = \langle \mathcal{V}_a, \mathcal{V}_q, \mathcal{E}_m \rangle$. Each node $i \in \mathcal{V}_a$ represents an agent, and each node $g \in \mathcal{V}_q$ represents a utility ($q_i$) or payoff ($q_{ij}$) function. Edges in $\mathcal{E}_m$ connect $g$ with the corresponding agent node(s). Message passing on this bipartite graph starts with sending messages from node $i \in \mathcal{V}_a$ to node $g \in \mathcal{V}_q$:

$$m_{i \to g}(a_i) = \sum_{h \in \mathcal{F}_i \setminus g} m_{h \to i}(a_i) + c_{ig}, \tag{2}$$

where $\mathcal{F}_i$ is the set of nodes connected to node $i$ in $\mathcal{V}_q$, and $c_{ig}$ is a normalizing factor preventing the value of messages from growing arbitrarily large. The message from node $g$ to node $i$ is:

$$m_{g \to i}(a_i) = \max_{\boldsymbol{a}_g \setminus a_i} \left[ q(\boldsymbol{a}_g) + \sum_{h \in \mathcal{V}_g \setminus i} m_{h \to g}(a_h) \right], \tag{3}$$

where $\mathcal{V}_g$ is the set of nodes connected to node $g$ in $\mathcal{V}_a$, $\boldsymbol{a}_g = \{a_h | h \in \mathcal{V}_g\}$, $\boldsymbol{a}_g \setminus a_i = \{a_h | h \in \mathcal{V}_g \setminus \{i\}\}$, and $q$ represents utility or payoff functions conditioned on $\boldsymbol{a}_g$. After several iterations of message passing, each agent $i$ can find its optimal action by calculating $a_i^* = \operatorname{argmax}_{a_i} \sum_{h \in \mathcal{F}_i} m_{h \to i}(a_i)$.

A drawback of Max-Sum or other message passing methods (e.g., max-plus (Pearl, 2014)) is that running them for each action selection through the whole system results in intensive computation and communication among agents, which is impractical for most applications with limited computational resources and communication bandwidth. In the following sections, we discuss how to solve this problem by learning sparse coordination graphs.

Previous works (Naderializadeh et al., 2020; Li et al., 2021) study soft versions of fully-connected coordination graphs based on attention mechanisms. Specifically, Li et al. (2021) uses graphs whose edge weights are learned by self-attention so that agents attend to observations of other agents differently. The information is used in local actors or a centralized critic. Naderializadeh et al. (2020) learns soft full graphs in a similar way, but the graph is used to mix local utilities conditioned on local action-observation history. Different from our work, these methods do not learn pairwise payoff functions, and the learned graphs are still fully-connected.

## 3 LEARNING CONTEXT-AWARE SPARSE GRAPHS

In this section, we introduce our methods for learning context-aware sparse graphs. We first introduce how we construct a sparse graph for effective action selection in Sec. 3.1. After that, we introduce our learning framework in Sec. 3.2. Although sparse graphs can reduce communication overhead, they raise the concern of learning instability. We discuss this problem and how to alleviate it in Sec. 3.3.

### 3.1 CONSTRUCT SPARSE GRAPHS

Action values, especially the pairwise payoff functions, contain much information about mutual influence between agents. Let's consider two agents $i$ and $j$. Intuitively, agent $i$ needs to coordinate its action selection with agent $j$ if agent $j$'s action exerts significant influence on the expected utility of agent $i$. For a fixed action $a_i$, $\operatorname{Var}_{a_j}[q_{ij}(\boldsymbol{\tau}_{ij}, \boldsymbol{a}_{ij})]$ can measure the influence of agent $j$ on the expected payoff. This intuition motivates us to use the **variance of payoff functions**

$$\zeta_{ij}^{q_{\mathrm{var}}} = \max_{a_i} \operatorname{Var}_{a_j}[q_{ij}(\boldsymbol{\tau}_{ij}, \boldsymbol{a}_{ij})], \tag{4}$$

as an indicator to construct sparse graphs. The maximization operator guarantees that the most affected action is considered. When $\zeta_{ij}^{q_{\mathrm{var}}}$ is large, the expected utility of agent $i$ fluctuates dramatically with the action of agent $j$, and they need to coordinate their actions. Therefore, with this measurement, to construct sparse coordination graphs, we can set a sparseness controlling constant $\lambda \in (0, 1)$ and select $\lambda |\mathcal{V}|(|\mathcal{V}| - 1)$ edges with the largest $\zeta_{ij}^{q_{\mathrm{var}}}$ values.

To justify this approach, we theoretically prove that, the smaller the value of $\zeta_{ij}^{q_{\mathrm{var}}}$ is, the more likely that the Max-Sum algorithm will select the same actions after removing the edge $(i, j)$.

**Proposition 1.** *For any two agents $i$, $j$ and the edge $e_{ij}$ connecting them in the coordination graph, after removing edge $e_{ij}$, greedy actions of agent $i$ and $j$ selected by the Max-Sum algorithm keep unchanged with a probability larger than*

$$\frac{2}{|A|} \left[ \frac{(\bar{m} - \min_{a_j} m(a_j))(\max_{a_j} m(a_j) - \bar{m})}{\left[ \zeta_{ij}^{q_{var}} + 2M^2 + 2\sqrt{M^2 \left( M^2 + \zeta_{ij}^{q_{var}} \right)} \right]^2} - 1 \right], \tag{5}$$

*where $m(a_j) = m_{e_{ij} \to j}(a_j)$, $\bar{m}$ is the average of $m(a_j)$, $M = \max_{a_j} \left[ \max_{a_i} r(a_i, a_j) - r(a_i, a_j) \right]$, and $r(a_i, a_j) = q(a_i, a_j) + m_{i \to e_{ij}}(a_j)$.*

Detailed proof can be found in Appendix A. The lower bound in Proposition 1 increases with a decreasing $\zeta_{ij}^{q_{var}}$. Therefore, edges with a smaller $\zeta_{ij}^{q_{var}}$ are less likely to influence the results of Max-Sum, justifying the way we construct sparse graphs.

### 3.2 LEARNING FRAMEWORK

Like conventional Q-learning, CASEC consists of two main components – learning value functions and selecting greedy actions. The difference is that these two steps are now carried out on dynamic and sparse coordination graphs.

In CASEC, agents learn a shared utility function $q_{\xi_u}(\cdot|\tau_i)$, parameterized by $\xi_u$, and a shared pairwise payoff function $q_{\xi_p}(\cdot|\boldsymbol{\tau}_{ij})$, parameterized by $\xi_p$. The global Q value function is estimated as:

$$Q_{tot}(\boldsymbol{\tau}, \boldsymbol{a}) = \frac{1}{|\mathcal{V}|} \sum_i q_{\xi_u}(a_i|\tau_i) + \frac{1}{|\mathcal{V}|(|\mathcal{V}| - 1)} \sum_{i \neq j} q_{\xi_p}(\boldsymbol{a}_{ij}|\boldsymbol{\tau}_{ij}), \tag{6}$$

which is updated by the TD loss:

$$\mathcal{L}_{TD}(\xi_u, \xi_p) = \mathbb{E}_{\mathcal{D}} \left[ \left( r + \gamma \hat{Q}_{tot}(\boldsymbol{\tau}', \text{Max-Sum}(q_{\hat{\xi}_u}, q_{\hat{\xi}_p})) - Q_{tot}(\boldsymbol{\tau}, \boldsymbol{a}) \right)^2 \right]. \tag{7}$$

Max-Sum$(\cdot, \cdot)$ is the greedy joint action selected by Max-Sum, $\hat{Q}_{tot}$ is a target network with parameters $\hat{\xi}_u$, $\hat{\xi}_p$ periodically copied from $Q_{tot}$, and the expectation is estimated with uniform samples from a replay buffer $\mathcal{D}$. Meanwhile, we also minimize a sparseness loss

$$\mathcal{L}_{\text{sparse}}^{q_{var}}(\xi_p) = \frac{1}{|\mathcal{V}|(|\mathcal{V}| - 1)|A|} \sum_{i \neq j} \sum_{a_i} \text{Var}_{a_j} \left[ q_{ij}(\boldsymbol{\tau}_{ij}, \boldsymbol{a}_{ij}) \right], \tag{8}$$

which is a regularization on $\zeta_{ij}^{q_{var}}$. Introducing a scaling factor $\lambda_{\text{sparse}} \in (0, 1]$ and minimizing $\mathcal{L}_{TD}(\xi_u, \xi_p) + \lambda_{\text{sparse}} \mathcal{L}_{\text{sparse}}^{q_{var}}(\xi_p)$ builds in inductive biases which favor minimized coordination graphs that would not sacrifice global returns.

Actions with the maximized value are selected for Q-learning and execution. In our framework, such action selections are finished by running Max-Sum on sparse graphs induced by $\zeta_{ij}^{q_{var}}$ (while we update Q functions on the full graph). Running Max-Sum requires the message passing through each node and edge. To speed up action selections, similar to previous work (Böhmer et al., 2020), we use multi-layer graph neural networks without parameters to process messages in a parallel manner.

### 3.3 STABILIZE LEARNING

One question with estimating $q_{ij}$ is that there are $|A| \times |A|$ action-pairs, each of which requires an output head in conventional deep Q networks. As only executed action-pairs are updated during Q-learning, parameters of many output heads remain unchanged for long stretches of time, resulting in estimation errors. Previous work (Böhmer et al., 2020) uses a low-rank approximation to reduce the number of output heads. However, it is largely unclear how to choose the best rank $K$ for different tasks, and still only $\frac{1}{|A|}$ of the output heads are selected in one Q-learning update.

This problem of estimation errors becomes especially problematic in CASEC, where building coordination graphs relies on the estimation of $q_{ij}$. A negative feedback loop is created because the built coordination graphs also affect the learning of $q_{ij}$. This loop renders learning unstable (Fig. 5).

We propose to solve this question and stabilize training by 1) periodically fixing the way we construct graphs via using the target payoff function to build graphs; and 2) accelerating the training of payoff function between target network updates to reduce the estimation errors via learning action representations.

Specifically, for 2), we propose to condition the utility and payoff functions on action representations to improve sample efficiency. We train an action encoder $f_{\xi_a}(a)$, whose input is the one-hot encoding of an action $a$ and output is its representation $z_a$. We adopt the technique introduced by Wang et al. (2021b) to train an effect-based action encoder. Specifically, action representation $z_a$, together with the current local observations, is used to predict the reward and observations at the next timestep. The prediction loss is minimized to update the action encoder $f_{\xi_a}(a)$. For more details, we refer readers to Appendix D. The action encoder is trained with few samples when learning begins and remains unchanged for the rest of the training process.

Using action representations, the utility and payoff functions can now be estimated as:

$$
\begin{aligned}
q_{\xi_u}(\tau_i, a_i) &= h_{\xi_u}(\tau_i)^{\mathrm{T}} z_{a_i}; \\
q_{\xi_p}(\boldsymbol{\tau}_{ij}, \boldsymbol{a}_{ij}) &= h_{\xi_p}(\boldsymbol{\tau}_{ij})^{\mathrm{T}}[z_{a_i}, z_{a_j}],
\end{aligned}
\tag{9}
$$

where $h$ includes a GRU (Cho et al., 2014) to process sequential input and output a vector with the same dimension as the corresponding action representation. $[\cdot, \cdot]$ means vector concatenation. Using Eq. 9, no matter which action is selected for execution, all parameters in the framework ($\xi_u$ and $\xi_p$) would be updated. The detailed network structure can be found in Appendix E.

## 4 MULTI-AGENT COORDINATION BENCHMARK

To evaluate our sparse graph learning algorithm, we collect classic coordination problems from the cooperative multi-agent learning literature, improve their difficulty, and classify them into different types. Then, 6 representative problems are selected and presented as a new benchmark called Multi-Agent COordination (MACO) challenge (Table 1).

At the first level, tasks are classified as factored and non-factored games, where factored games present an explicit decomposition of global rewards. Factored games are further categorized according to two properties – whether the task requires pairwise or higher-order coordination, and whether the underlying coordination relationships change temporally. For non-factored games, we divide them into two classes by whether the task characterizes static coordination relationships among agents. To better test

Table 1: Multi-agent coordination benchmark.

| Task | Factored | Pairwise | Dynamic | # Agents |
|---|---|---|---|---|
| Aloha | $\checkmark$ | $\checkmark$ | | 10 |
| Pursuit | $\checkmark$ | $\checkmark$ | $\checkmark$ | 10 |
| Hallway | $\checkmark$ | | | 12 |
| Sensor | $\checkmark$ | | $\checkmark$ | 15 |
| Gather | | – | | 5 |
| Disperse | | – | $\checkmark$ | 12 |

the performance of different methods, we increase the difficulty of the included problems by extending stateless games to temporally extended settings (`Gather` and `Disperse`), complicating the reward function (`Pursuit`), or increasing the number of agents (`Aloha` and `Hallway`). We now briefly describe the setting of each game. For detailed description, we refer readers to Appendix B.

`Aloha` (Hansen et al., 2004; Oliehoek, 2010) consists of 10 islands in a $2 \times 5$ array. Each island has a backlog of messages to send. They send one message or not at each timestep. When two neighboring agents send simultaneously, messages collide and have to be resent. Islands start with 1 package in the backlog. At each timestep, with a probability $0.6$ a new packet arrives if the maximum backlog (5) has not been reached. Each agent observes its position and the number of packages in its backlog. Agents receive a global reward of $0.1$ for each successful transmission, and $-10$ for a collision.

`Pursuit`, also called `Predator and Prey`, is a classic coordination problem (Benda, 1986; Stone & Veloso, 2000; Son et al., 2019). Ten agents (predators) roam a $10 \times 10$ map populated with 5 random walking preys for 50 environment steps. One prey is captured if two agents catch it simultaneously, after which the catching agents and the prey are removed from the map, resulting in a team reward

Figure 1: **Left**: Learning curves (return and the number of successfully scanned targets) of CASEC and DCG on `Sensor`. **Middle**: The influence of graph sparseness on performance (return and the number of scanned targets). Here we show the case of the best seed. The plot of other seeds can be found in Fig. 21. **Right**: An example coordination graph learned by our method.

of $1$. If only one agent tries to catch the prey, the prey would not be captured and the agents will be punished. We consider a challenging version of `Pursuit` by setting the punishment to 1, which is the same as the reward obtained by a successful catch.

`Hallway` (Wang et al., 2020) is a multi-chain Dec-POMDP. We increase the difficulty of `Hallway` by introducing more agents and grouping them (Fig. 6). One agent randomly spawns at a state in each chain. Agents can observe their own position and choose to move left, move right, or keep still at each timestep. Agents win with a global reward of 1 if they arrive at state $g$ simultaneously with other agents in the same group. If $n_g > 1$ groups attempt to move to $g$ at the same timestep, they keep motionless and agents receive a global punishment of $-0.5 * n_g$.

`Sensor` has been extensively studied (Lesser et al., 2012; Zhang & Lesser, 2011). We consider 15 sensors in a $3 \times 5$ matrix. Sensors can scan the eight nearby points. Each scan induces a cost of -1, and agents can do `noop` to save the cost. Three targets wander randomly in the gird. If $k \geq 2$ sensors scan a target simultaneously, the system gets a constant reward of 3, which is independent of the number of sensors. Agents can observe the id and position of targets nearby.

`Gather` is an extension of the `Climb` Game (Wei & Luke, 2016). In `Climb` Game, each agent has three actions: $A = \{a_0, a_1, a_2\}$. Action $a_0$ yields no reward (0) if only some agents choose it, but a high reward (10) if all choose it. Otherwise, if no agent chooses action $a_0$, a reward $5$ is obtained. We increase the difficulty of this game by making it temporally extended and introducing stochasticity. Actions are no longer atomic, and agents need to learn policies to realize these actions by navigating to goals $g_1$, $g_2$ and $g_3$ (Fig. 7). Moreover, for each episode, one of $g_1$, $g_2$ and $g_3$ is randomly selected as the optimal goal (corresponding to $a_0$ in the original game).

`Disperse` consists of 12 agents. At each timestep, agents can choose to work at one of 4 hospitals by selecting an action in $A = \{a_0, a_1, a_2, a_3\}$. At timestep $t$, hospital $j$ needs $x_t^j$ agents for the next timestep. One hospital is randomly selected and its $x_t^j$ is a positive number, while the need of other hospitals is 0. If $y_{t+1}^j < x_t^j$ agents go to the selected hospital, the whole team would be punished $y_{t+1}^j - x_t^j$. Agents observe the local hospital's id and its need for the next timestep.

## 5 CASE STUDY: LEARNING SPARSE GRAPHS ON `Sensor`

We are particularly interested in the dynamics and results of sparse graph learning. Therefore, we carry out a case study on `Sensor`. When training CASEC on this task, we select $10\%$ edges with largest $\zeta_{ij}^{q_{\text{var}}}$ values to construct sparse graphs.

**Interpretable sparse coordination graphs.** In Fig. 1 right, we show a screenshot of the game with the learned coordination graph at a certain timestep. We can observe that all edges in the learned graph involve agents around the targets. Let's see the case of `agent 8`. The action proposed by the individual utility function of `agent 8` is to scan `target 1`. After coordinating its action with other agents, `agent 8` changes its action selection and scans target `target 2`, resulting in an optimal solution for the given configuration. This result is in line with our theoretical analysis in Sec. 3.1. The most important edges can be characterized by a large $\zeta$ value.

**Influence of graph sparseness on performance.** It is worth noting that with fewer edges in the coordination graph, CASEC has better performance than DCG on `Sensor` (Fig. 1 left, where the

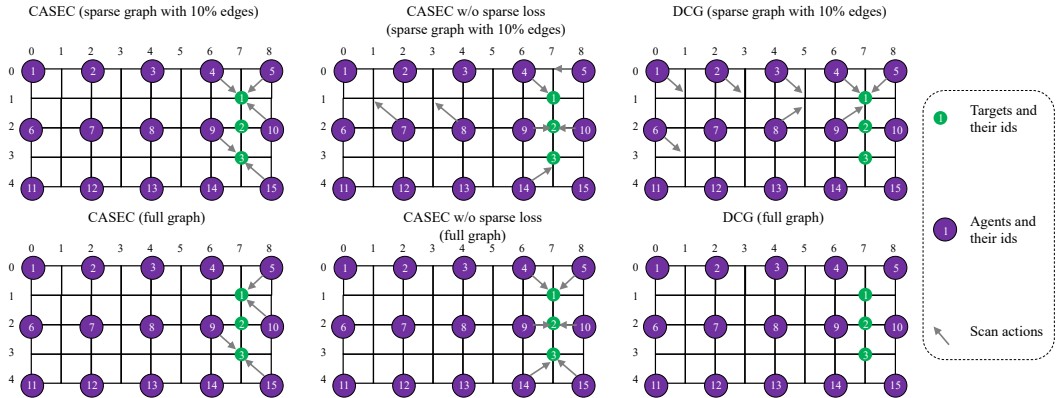

Figure 2: Coordination graphs learned by different methods on `Sensor`.

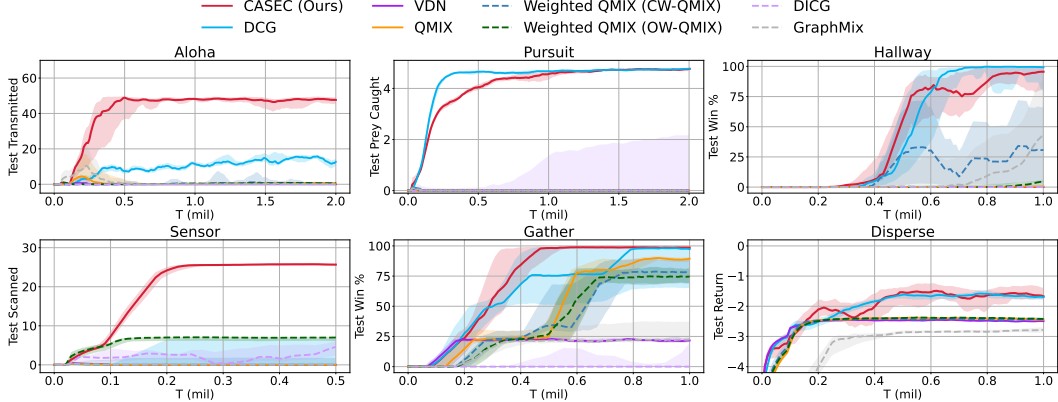

Figure 3: Performance comparison with baselines on the MACO benchmark.

median performance and 25%-75% percentiles are shown). This observation may be counter-intuitive at the first glance. To study this problem, we load the model after convergence learned by CASEC and DCG, gradually remove edges from the full graph in the ascending order of $\zeta_{ij}^{q_{\text{var}}}$, and check the change of scanned targets and the obtained reward. Results are shown in Fig. 1 middle and Fig. 21.

It can be observed that the performance of DCG (the number of scanned targets) does not change with the number of edges. In another word, only the individual utility function contributes to scanning targets. Screenshots shown in Fig. 2 (right column) align with this observation. With more edges, DCG makes a less optimal decision: `agent 4, 5`, and `9` no longer scan target 1.

In contrast, the performance of CASEC grows with more edges in the coordination graph. By referring to Fig. 2 (left column), we can conclude that CASEC first selects edges that help agents scan more targets, and then selects edges that can eliminate useless scan actions. These results demonstrate that our method can distinguish the most important edges on `Sensor`.

We also study the influence of the sparseness loss (Eq. 8). As shown in Fig. 1 middle, CASEC without the sparseness loss consistently gets fewer rewards than CASEC. For example, target 1 and 3 are not captured in the case shown in Fig. 2 (middle column) as only one agent scans them. These results highlight the function of the sparseness loss.

## 6 EXPERIMENTS

In this section, we design experiments to answer the following questions: (1) How much communication can be saved by our method? How does communication threshold influence performance on factored and non-factored games? (Sec. 6.1) (2) How does our method compare to state-of-the-art multi-agent learning methods? (Sec. 6.2, 6.3) (3) Is our method efficient in settings with larger

action-observation spaces? (Sec. 6.3) For results in this section, we show the median performance with 8 random seeds as well as the 25-75% percentiles.

## 6.1 GRAPH SPARSENESS

An important advantage of learning sparse coordination graphs is reduced communication costs. The complexity of running Max-Sum for each action selection is $\mathcal{O}\left(k\left(|\mathcal{V}||A| + |\mathcal{E}||A|^2\right)\right)$, where $k$ is the number of iterations of message passing. Sparse graphs cut down communication costs by reducing the number of edges.

We carry out a grid search to find the communication threshold under which sparse graphs have the best performance. We find that most implementations of dynamically sparse graphs require

Table 2: Percentage of communication saved for each task.

| Aloha | Pursuit | Hallway |
| --- | --- | --- |
| 80.0% | 70.0% | 50.0% |
| Sensor | Gather | Disperse |
| 90.0% | 30.0% | 60.0% |

similar numbers of edges to prevent performance from dropping significantly. In Table 2, we show the communication cut rates we use when benchmarking our method. Generally speaking, non-factored games typically require more messages than factored games, while, for most tasks, at least $50\%$ messages can be saved without sacrificing the learning performance.

**Communication threshold vs. performance** In Fig. 4, we show the performance of our method under different communication thresholds which control the sparseness of edges. We can observe that, on the factored game `Sensor`, performance first grows then drops when more edges are included in the coordination graphs. These observations are in line with the fact that sparse graphs can outperform complete graphs and fully-decomposed value functions on this task. In contrast, for the non-factored game `Gather`, performance stabilizes beyond a certain threshold. Non-factored games usually involve complex coordination relationships, and denser topologies are suitable for this type of questions.

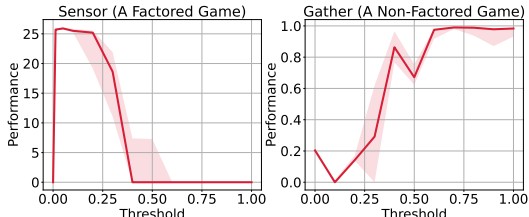

Figure 4: The influence of graph sparseness (1.0 represents complete graphs) on the performance on factored games (`Sensor`, left) and non-factored games (`Gather`, right).

## 6.2 MACO: MULTI-AGENT COORDINATION BENCHMARK

We compare our method with state-of-the-art fully-decomposed value-based methods (VDN (Sunehag et al., 2018), QMIX (Rashid et al., 2018), and Weighted QMIX (Rashid et al., 2020)), coordination graph learning method (DCG (Böhmer et al., 2020)), and attentional graph learning methods (DICG (Li et al., 2021) and GraphMIX (Naderializadeh et al., 2020)) on MACO (Fig. 3). Since the number of actions is not very large in MACO, we do not use action representations when estimating the utility and payoff function for CASEC.

We can see that our method significantly outperforms fully-decomposed value-based methods. The reason is that fully-decomposed methods suffer from the *relative overgeneralization* issue and miscoordination problems in partially observable environments with stochasticity. For example, on task `Pursuit` (Benda, 1986), if more than one agent catches one prey simultaneously, these agents will be rewarded 1. However, if only one agent catches prey, it fails and gets a punishment of -1. For an agent with a limited sight range, the reward it obtains when taking the same action (catching a prey) under the same local observation depends on the actions of other agents and changes dramatically. This is the relative overgeneralization problem. Another example is `Hallway` (Wang et al., 2020), where several agents need to reach a goal state simultaneously without knowing each other's location. Fully-decomposed methods cannot solve this problem if the initial positions of agents are stochastic.

For DCG, we use its default settings of complete graphs and no low-rank approximation. We observe that DCG is less effective on tasks characterized by sparse coordination interdependence like `Sensor`. We hypothesize this is because coordinating actions with all other agents requires the shared estimator to express payoff functions of most agent pairs accurately enough, which needs more samples to learn, hurting the performance of DCG on loosely coupled tasks.

Figure 5: Performance and TD errors compared to baselines and ablations on the SMAC benchmark.

## 6.3 STARCRAFT II MICROMANAGEMENT BENCHMARK

We compare our method against the state-of-the-art coordination graph learning method (DCG (Böhmer et al., 2020)) and fully decomposed value-based MARL algorithms (VDN (Sunehag et al., 2018), QMIX (Rashid et al., 2018)). For CASEC, we use action representations to estimate the payoff function. We train the action encoder for $50k$ samples and keep action representations unchanged afterward. In Fig. 5, we show results on 5m_vs_6m and MMM2. Detailed hyperparameter settings of our method can be found in Appendix E.

For DCG, we use its default settings, including a low-rank approximation for learning the payoff function. We can see that CASEC outperforms DCG by a large margin. The result proves that sparse coordination graphs provide better scalability to large action-observation spaces than dense and static graphs. In DCG's defense, low-rank approximation still induces large estimation errors. We replace low-rank approximation with action representations and find that DCG (*Full (action repr.)*) achieves similar performance to CASEC after 5M steps, but CASEC is still more sample-efficient. Moreover, taking advantage of higher-order value decomposition, CASEC is able to represent more complex coordination dynamics than fully decomposed value functions and thus performs better.

**Ablation study** Our method is characterized by two contributions: context-aware sparse topologies and action representations for learning the utility and payoff function. In this section, we design ablations to show their contributions.

The effect of sparse topologies can be observed by comparing CASEC to *Full (action repr.)*, which is the same as CASEC other than using complete coordination graphs. We observe that sparse graphs enjoy better sample efficiency than full graphs, and the advantage becomes less obvious as more samples are collected. This observation indicates that sparse graphs introduce inductive biases that can accelerate training, and their representational capacity is similar to that of full graphs.

From the comparison between CASEC to CASEC using conventional Q networks (*w/o action repr.*), we can see that using action representations can significantly stabilize learning. For example, learning diverges on 5m_vs_6m without action representations. As analyzed before, this is because a negative feedback loop is created between the inaccurate payoff function and coordination graphs.

To further consolidate that action representations can reduce the estimation errors and thus alleviate learning oscillation as discussed in Sec. 3.3, we visualize the TD errors of CASEC and ablations during training in Fig. 5 right. We can see that action representations can dramatically reduce the TD errors. For comparison, the low-rank approximation can also reduce the TD errors, but much less significantly. Smaller TD errors prove that action representations provide better estimations of the value function, and learning with sparse graphs can thus be stabilized (Fig. 5 left).

## 7 CONCLUSION

We study how to learn dynamic sparse coordination graphs, which is a long-standing problem in cooperative MARL. We propose a specific implementation and theoretically justify it. Empirically, we evaluate the proposed method on a new multi-agent coordination benchmark. Moreover, we equip our method with action representations to improve the sample efficiency of payoff learning and stabilize training. We show that sparse and adaptive topologies can largely reduce communication overhead as well as improve the performance of coordination graphs. We expect our work to extend MARL to more realistic tasks with complex coordination dynamics.

One limitation of our method is that the learned sparse graphs are not always cycle-free. Since the Max-Sum algorithm guarantees optimality only on acyclic graphs, our method may select sub-optimal actions. In Appendix F, we study this problem in depth.

Another limitation is that we fix the communication threshold when training. It is an important question how to automatically and accurately find the minimum threshold that can guarantee the learning performance. In Appendix I, we study two ways to adaptively select the threshold.

**Reproducibility**    The source code for all the experiments along with a README file with instructions on how to run these experiments is attached in the supplementary material. In addition, the settings and parameters for all models and algorithms mentioned in the experiment section are detailed in Appendix E.

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

# A    MATHEMATICAL PROOF

In this section, we provide proof to Proposition 1.

Without loss of generality, we consider two agents $1$ and $2$ and the edge between them $(1, 2)$. We prove our idea by comparing the action selection of agent 2 before and after removing edge $(1, 2)$. In the following proof, we use $i$ ($i = 1, 2$) to denote agent $i$ and $e$ to denote edge $(1, 2)$.

Action of agent 2 is determined by

$$a_2^* = \arg\max_{a_2} \sum_{h \in \mathcal{F}_2} m_{h \to 2}(a_2) \tag{10}$$

$$= \arg\max_{a_2} \left[ m_{e \to 2}(a_2) + \sum_{h \in \mathcal{F}_2/\{e\}} m_{h \to 2}(a_2) \right], \tag{11}$$

and we first see the influence of $m_{e \to 2}(a_2)$ on $a_2^*$. For clarity, we use $m(a_2)$ to denote $m_{e \to 2}(a_2)$ and $l(a_2)$ to denote $\sum_{h \in \mathcal{F}_2/\{e\}} m_{h \to 2}(a_2)$. We are interested in whether $\arg\max_{a_2} l(a_2) = \arg\max_{a_2} [m(a_2) + l(a_2)]$. The probability of this event holds if the following inequality holds:

$$\text{Range}\,[m(a_2)] \le \min_{a_2 \ne a_2^j} (l(a_2^j) - l(a_2)), \tag{12}$$

where $\text{Range}(\boldsymbol{x})$ denotes the largest elements in vector $\boldsymbol{x}$ minus the smallest one and $a_2^j = \arg\max_{a_2} l(a_2)$. We rewrite Eq. 12 and obtain

$$Pr\left( \text{Range}\,[m(a_2)] \le \min_{a_2 \ne a_2^j} (l(a_2^j) - l(a_2)) \right) \tag{13}$$

$$= Pr\left( \min_{a_2} m(a_2) < m(a_2) < \min_{a_2} m(a_2) + \min_{a_2 \ne a_2^j} (l(a_2^j) - l(a_2)) \right). \tag{14}$$

According to the Asymmetric two-sided Chebyshev's inequality (Mitzenmacher & Upfal, 2017), we get a lower bound of this probability:

$$4 \frac{(\bar{m} - \min_{a_2} m(a_2))(\max_{a_2} m(a_2) - \bar{m}) - \sigma^2}{\left[ \min_{a_2 \ne a_2^j} (l(a_2^j) - l(a_2)) \right]^2}, \tag{15}$$

where $\sigma$ is the variance of $m(a_2)$, and $\bar{m}$ is the average of $m(a_2)$.

Suppose that we take $|A|$ actions independently. According to the von Szokefalvi Nagy inequality (Nagy, 1918), we can further get the lower bound as follows:

$$4 \frac{(\bar{m} - \min_{a_2} m(a_2))(\max_{a_2} m(a_2) - \bar{m}) - \sigma^2}{\left[ \min_{a_2 \ne a_2^j} (l(a_2^j) - l(a_2)) \right]^2} \ge 4 \frac{(\bar{m} - \min_{a_2} m(a_2))(\max_{a_2} m(a_2) - \bar{m}) - \sigma^2}{2|A|\sigma^2}$$

$$= \frac{2}{|A|} \left[ \frac{(\bar{m} - \min_{a_2} m(a_2))(\max_{a_2} m(a_2) - \bar{m})}{\sigma^2} - 1 \right]. \tag{16}$$

Note that

$$m(a_2) = \max_{a_1} [q(a_1, a_2) + m_{1 \to e}(a_1)], \tag{17}$$

and we are interested in $q(a_1, a_2)$. We now study the relationship between $m(a_2)$ and $\max_{a_1} [q(a_1, a_2)]$. For clarity, we use $r(a_1, a_2)$ to denote $q(a_1, a_2) + m_{1 \to e}(a_1)$, and $r(a_1^{i_2}, a_k)$ to denote $\max_{a_1} r(a_1, a_2)$. Then we have $\text{Var}_{a_2} \max_{a_1} r(a_1, a_2) = \text{Var}_{a_2} r(a_1^{i_2}, a_2)$.

For a given $a_2$, we have

$$\text{Var}_{a_2} r(a_1, a_2) = \text{Var}_{a_2} \left[ r(a_1^{i_2}, a_2) - s_2 \right]. \tag{18}$$

Here $s_2 \geq 0, \forall a_1$, because $i_2 = \arg\max_i r(a_1, a_2)$.

Since

$$\text{Var}_{a_2}\left[r(a_1^{i_2}, a_2) - s_2\right] \tag{19}$$

$$=\text{Var}_{a_2}\left[r(a_1^{i_2}, a_2)\right] + \text{Var}\left[s_2\right] - 2\text{Cov}(r(a_1^{i_2}, a_2), s_2) \tag{20}$$

and

$$\text{Cov}(r(a_1^{i_2}, a_2), s_2) \leq \sqrt{\text{Var}_{a_2}\left[r(a_1^{i_2}, a_2)\right] \text{Var}\left[s_2\right]}, \tag{21}$$

it follows that

$$\text{Var}_{a_2}\left[r(a_1^{i_2}, a_2) - s_2\right] \tag{22}$$

$$\geq\text{Var}_{a_2}\left[r(a_1^{i_2}, a_2)\right] - 2\sqrt{\text{Var}_{a_2}\left[r(a_1^{i_2}, a_2)\right] \text{Var}\left[s_2\right]}. \tag{23}$$

Thus,

$$\zeta_{12}\left[r(a_1, a_2)\right] = \max_{a_1} \text{Var}_{a_2}\left[r(a_1, a_2)\right] \geq \text{Var}_{a_2} \max_{a_1}\left[r(a_1, a_2)\right] - 2\sqrt{\text{Var}_{a_2}\left[r(a_1^{i_2}, a_2)\right] \text{Var}\left[s_2\right]}. \tag{24}$$

Observing that $\zeta_{12}\left[r(a_1, a_2)\right] = \max_{a_1} \text{Var}_{a_2}\left[r(a_1, a_2)\right] = \max_{a_1} \text{Var}_{a_2}\left[q(a_1, a_2)\right] = \zeta_{12}\left[q(a_1, a_2)\right]$, we have

$$\sigma \leq \zeta_{12}\left[q(a_1, a_2)\right] + 2\sqrt{\text{Var}_{a_2}\left[r(a_1^{i_2}, a_2)\right] \text{Var}\left[s_2\right]}$$
$$= \zeta_{12}\left[q(a_1, a_2)\right] + 2\sqrt{\sigma S}, \tag{25}$$

where $\sigma = \text{Var}_{a_2} \max_{a_1}\left[r(a_1, a_2)\right]$ and $\text{Var}\left[s_2\right] = S$. According to the fixed-point theorem, the term $\sigma$ satisfies $\zeta_{12}\left[q(a_1, a_2)\right] + 2\sqrt{\sigma S} = \sigma$. We can solve this quadratic form and get $\sigma = \zeta_{12}\left[q(a_1, a_2)\right] + 2S \pm 2\sqrt{S\left(S + \zeta_{12}\left[q(a_1, a_2)\right]\right)}$. Because the $\sigma$ term is larger than $\zeta_{12}\left[q(a_1, a_2)\right] + 2S$, we get $\sigma = \zeta_{12}\left[q(a_1, a_2)\right] + 2S + 2\sqrt{S\left(S + \zeta_{12}\left[q(a_1, a_2)\right]\right)}$. By inserting this inequality to the lower bound (Eq. 16), we get a lower bound related to $q(a_1, a_2)$:

$$\frac{2}{|A|}\left[\frac{(\bar{m} - \min_{a_2} m(a_2))(\max_{a_2} m(a_2) - \bar{m})}{\left[\zeta_{12}\left[q(a_1, a_2)\right] + 2S + 2\sqrt{S\left(S + \zeta_{12}\left[q(a_1, a_2)\right]\right)}\right]^2} - 1\right]. \tag{26}$$

When a vector $x$ is larger than 0 and the the cardinality of $x$ is $n$, we have: $\text{Var}(x) = \frac{1}{n}\sum_{i=1}^{n}(x_i - \frac{1}{n}\sum_{i=1}^{n} x_i)^2 \leq \frac{1}{n}\sum_{i=1}^{n} x_i^2 \leq \max_i x_i^2$. Thus we can further get the following bound:

$$S = \text{Var}_{a_2}\left[\max_{a_1} r(a_1, a_2) - r(a_1, a_2)\right]$$
$$\leq max_{a_2}^2\left[\max_{a_1} r(a_1, a_2) - r(a_1, a_2)\right]. \tag{27}$$

Let $M = \max_{a_2}\left[\max_{a_1} r(a_1, a_2) - r(a_1, a_2)\right]$, thus we have $S \leq M^2$. We have $\zeta_{12}\left[q(a_1, a_2)\right] = \zeta_{12}^{q_{\text{var}}}$ by the definition (Eq. 4) and can get the final lower bound:

$$\frac{2}{|A|}\left[\frac{(\bar{m} - \min_{a_2} m(a_2))(\max_{a_2} m(a_2) - \bar{m})}{\left[\zeta_{12}^{q_{\text{var}}} + 2M^2 + 2\sqrt{M^2\left(M^2 + \zeta_{12}^{q_{\text{var}}}\right)}\right]^2} - 1\right]. \tag{28}$$

# B  MACO: MULTI-AGENT COORDINATION BENCHMARK

In this paper, we study how to learn context-aware sparse coordination graphs. For this purpose, we propose a new Multi-Agent COordination (MACO) benchmark (Table 1) to evaluate different implementations and benchmark our method. This benchmark collects classic coordination problems in the literature of cooperative multi-agent learning, increases their difficulty, and classifies them into different types. We now describe the detailed settings of tasks in the MACO benchmark.

B.1 TASK SETTINGS

**Factored Games** are characterized by a clear factorization of global rewards. We further classify factored games into 4 categories according to whether coordination dependency is pairwise and whether the underlying coordination graph is dynamic (Table 1).

`Aloha` (Oliehoek (2010), also similar to the Broadcast Channel benchmark problem proposed by Hansen et al. (2004)) consists of 10 islands, each equipped with a radio tower to transmit messages to its residents. Each island has a backlog of messages that it needs to send, and agents can choose to send one message or not at each timestep. Due to the proximity of islands, communications from adjacent islands interfere. This means that when two neighboring agents attempt to send simultaneously, a collision occurs and the messages have to be resent. Each island starts with $1$ package in its backlog. At each timestep, with probability $0.6$ a new packet arrives if the maximum backlog (set to $5$) has not been reached. Each agent observes its position and the number of packages in its backlog. A global reward of $0.1$ is received by the system for each successful transmission, while punishment of $-10$ is induced if the transmission leads to a collision.

`Pursuit`, also called `Predator and Prey`, is a classic coordination problem (Benda, 1986; Stone & Veloso, 2000; Son et al., 2019). In this game, ten agents (predators) roam a $10 \times 10$ map populated with 5 random walking preys for 50 environment steps. Based on the partial observation of any adjacent prey and other predators, agents choose to move in four directions, keep motionless, or catch prey (specified by its id). One prey is captured if two agents catch it simultaneously, after which the catching agents and the prey are removed from the map, resulting in a team reward of $1$. However, if only one agent tries to catch the prey, the prey would not be captured and the agents will be punished. The difficulty of `Pursuit` is largely decided by the relative scale of the punishment compared to the catching reward (Böhmer et al., 2020), because a large punishment exacerbates the relative overgeneralization pathology. In the MACO benchmark, we consider a challenging version of `Pursuit` by setting the punishment to 1, which is the same as the reward obtained by a successful catch.

`Hallway` (Wang et al., 2020) is a multi-chain Dec-POMDP whose stochasticity and partial observability lead to fully-decomposed value functions learning sub-optimal strategies. In the MACO benchmark, we increase the difficulty of `Hallway` by introducing more agents and grouping them (Fig. 6). One agent randomly spawns at a state in each chain. Agents can observe its own position and choose to move left, move right, or keep still at each timestep. Agents win if they arrive at state $g$ simultaneously with other agents in the same group. In Fig. 6, different groups are drawn in different colors.

Each winning group induces a global reward of $1$. Otherwise, if any agent arrives at $g$ earlier than others, the system receives no reward and all agents in that group would be removed from the game. If $n_g > 1$ groups attempt to move to $g$ at the same timestep, they keep motionless and agents receive a global punishment of $-0.5 * n_g$. The horizon is set to $max_i\ l_i + 10$ to avoid an infinite loop, where $l_i$ is the length of chain $i$.

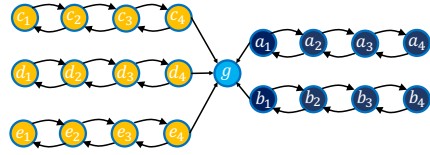

Figure 6: Task `Hallway` (Wang et al., 2020). To increase the difficulty of the game, we consider a multi-group version. Different colors represent different groups.

`Sensor` (Fig. 3 in the main text) has been extensively studied in cooperative multi-agent learning (Lesser et al., 2012; Zhang & Lesser, 2011). We consider 15 sensors arranged in a 3 by 5 matrix. Each sensor is controlled by an agent and can scan the eight nearby points. Each scan induces a cost of -1, and agents can choose `noop` to save the cost. Three targets wander randomly in the gird. If $k \geq 2$ sensors scan a target simultaneously, the system gets a constant reward of 3, which is independent of the number of sensor. Agents can observe the id and position of targets nearby.

**Non-factored games** do not present an explicit decomposition of global rewards. We classify non-factored games according to whether the game can be solved by a static (sparse) coordination graph in a single episode.

`Gather` is an extension of the `Climb Game` (Wei & Luke, 2016). In `Climb Game`, each agent has three actions: $A = \{a_0, a_1, a_2\}$. Action $a_0$ yields no reward if only some agents choose it, but a high reward if all choose it. The other two actions are sub-optimal actions but can induce a positive reward

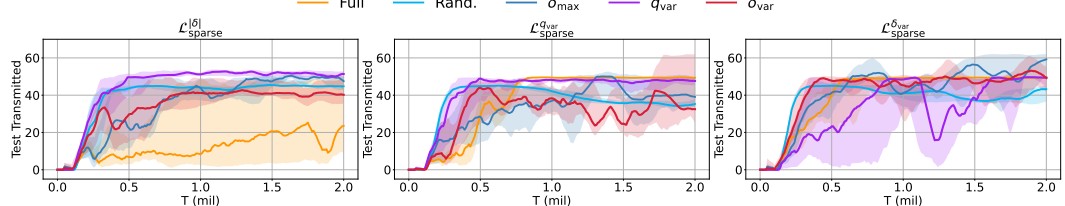

Figure 8: Performance of different implementations on `Aloha`. Different colors indicate different topologies. Performance of different losses is shown in different sub-figures.

without requiring precise coordination:

$$r(\boldsymbol{a}) = \begin{cases} 10 & \#a_0 = n, \\ 0 & 0 < \#a_0 < n, \\ 5 & \text{otherwise.} \end{cases} \tag{29}$$

We increase the difficulty of this game by making it temporally extended and introducing stochasticity. We consider three actions. Actions are no longer atomic, and agents need to learn policies to realize these actions by navigating to goals $g_1$, $g_2$ and $g_3$ (Fig. 7).

Moreover, for each episode, one of $g_1$, $g_2$ and $g_3$ is randomly selected as the optimal goal (corresponding to $a_0$ in Eq. 29). Agents spawn randomly, and only agents initialized near the optimal goal know which goal is optimal. Agents need to simultaneously arrive at a goal state to get any reward. If all agents are at the optimal goal state, they get a high reward of 10. If all of them are at other goal states, they would be rewarded 5. The minimum reward would be received if only some agents gather at the optimal goal. We further increase the difficulty by setting this reward to $-5$. It is worth noting that, for any single episode, `Gather` can be solved using a static and sparse coordination graph – for example, agents can collectively coordinate with an agent who knows the optimal goal.



Figure 7: Task `Gather`. To increase the difficulty of this game, we consider a temporally extended version and introduce stochasticity.

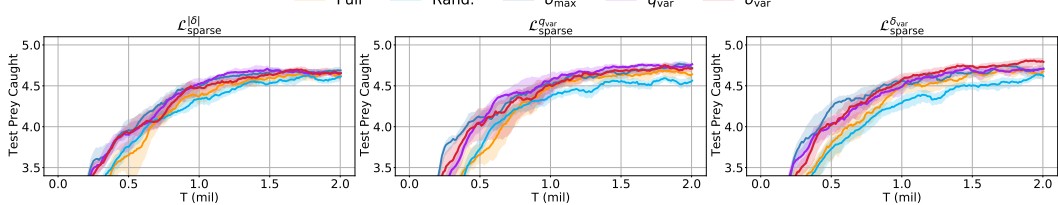

Figure 9: Performance of different implementations on `Pursuit`. Different colors indicate different topologies. Performance of different losses is shown in different sub-figures.

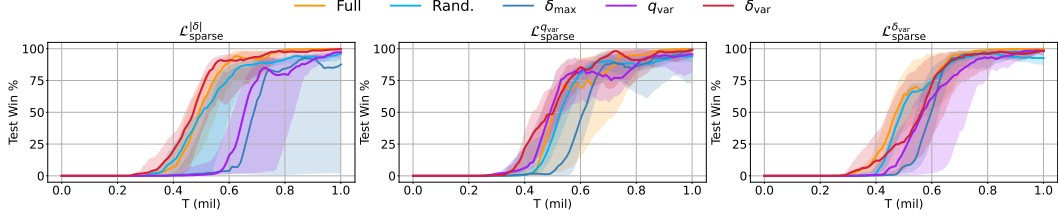

Figure 10: Performance of different implementations on `Hallway`. Different colors indicate different topologies. Performance of different losses is shown in different sub-figures.

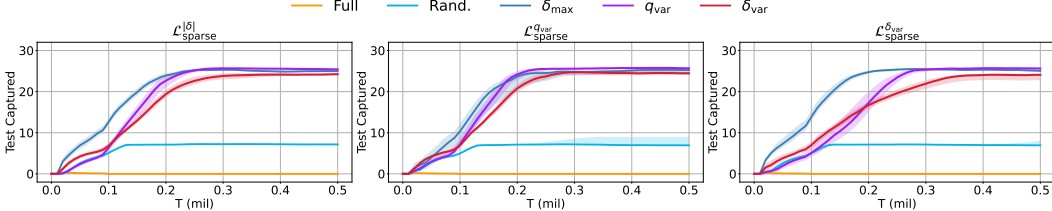

Figure 11: Performance of different implementations on `Sensor`. Different colors indicate different topologies. Performance of different losses is shown in different sub-figures.

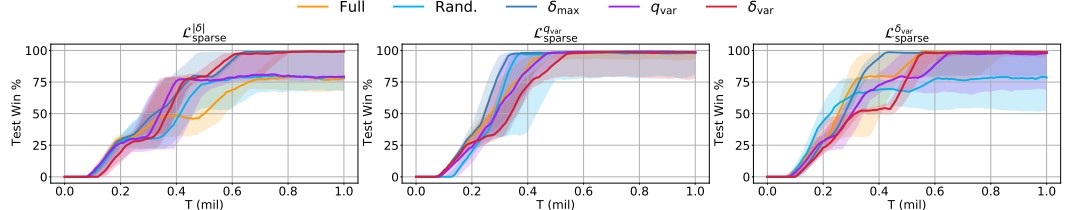

Figure 12: Performance of different implementations on `Gather`. Different colors indicate different topologies. Performance of different losses is shown in different sub-figures.

`Disperse` consists of 12 agents. At each timestep, agents can choose to work at one of 4 hospitals by selecting an action in $A = \{a_0, a_1, a_2, a_3\}$. At timestep $t$, hospital $j$ needs $x_t^j$ agents for the next timestep. One hospital is randomly selected and its $x_t^j$ is a positive number, while the need of other hospitals is 0. If $y_{t+1}^j < x_t^j$ agents go to the selected hospital, the whole team would be punished $y_{t+1}^j - x_t^j$. Agents observe the local hospital's id and its need for the next timestep.

## B.2  OTHER POSSIBLE IMPLEMENTS AND PERFORMANCE COMPARISON

With this benchmark in hand, we are now able to evaluate our method for constructing sparse graphs. We compare our method with the following approaches.

**Maximum utility difference**  $q_i$ (or $q_j$) is the expected utility agent $i$ (or $j$) can get without the awareness of actions of other agents. After specifying the action of agent $j$ or $i$, the joint expected utility changes to $q_{ij}$. Thus the measurement

$$\zeta_{ij}^{\delta_{\max}} = \max_{\boldsymbol{a}_{ij}} |\delta_{ij}(\boldsymbol{\tau}_{ij}, \boldsymbol{a}_{ij})| \tag{30}$$

can describe the mutual influence between agent $i$ and $j$. Here

$$\delta_{ij}(\boldsymbol{\tau}_{ij}, \boldsymbol{a}_{ij}) = q_{ij}(\boldsymbol{\tau}_{ij}, \boldsymbol{a}_{ij}) - q_i(\tau_i, a_i) - q_j(\tau_j, a_j) \tag{31}$$

is the ***utility difference function***.

We use a maximization operator here because two agents need to coordinate with each other if such coordination significantly affects the probability of selecting at least one action pair.

**Variance of utility difference**  As discussed before, the value of utility difference $\delta_{ij}$ and variance of payoff functions can measure the mutual influence between agent $i$ and $j$. In this way, the variance of $\delta_{ij}$ serves as a second-order measurement, and we can use

$$\zeta_{ij}^{\delta_{\mathrm{var}}} = \max_{a_i} \mathrm{Var}_{a_j} \left[ \delta_{ij}(\boldsymbol{\tau}_{ij}, \boldsymbol{a}_{ij}) \right] \tag{32}$$

to rank the necessity of coordination relationships between agents. Again we use the maximization operation to base the measurement on the most influenced action.

For these three measurements (Eq. 8, 30, and 32), the larger value of $\zeta_{ij}$ is, the more important the edge $(i, j)$ is. For example, when $\zeta_{ij}^{q_{\mathrm{var}}} = \max_{a_i} \mathrm{Var}_{a_j} \left[ q_{ij}(\boldsymbol{\tau}_{ij}, \boldsymbol{a}_{ij}) \right]$ is large, the expected utility of

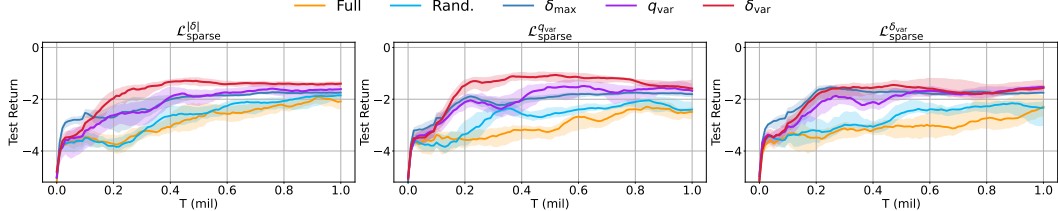

Figure 13: Performance of different implementations on `Disperse`. Different colors indicate different topologies. Performance of different losses is shown in different sub-figures.

agent $i$ fluctuates dramatically with the action of agent $j$, and they need to coordinate their actions. Therefore, with these measurements, to learn sparse coordination graphs, we can set a sparseness controlling constant $\lambda \in (0, 1)$ and select $\lambda|\mathcal{V}|(|\mathcal{V}| - 1)$ edges with the largest $\zeta_{ij}$ values. To make the measurements more accurate in edge selection, we minimize the following losses for the two measurements, respectively:

$$\mathcal{L}_{\text{sparse}}^{|\delta|} = \frac{1}{|\mathcal{V}|(|\mathcal{V}| - 1)|A|^2} \sum_{i \neq j} \sum_{a_i, a_j} |\delta_{ij}(\boldsymbol{\tau}_{ij}, \boldsymbol{a}_{ij})|; \tag{33}$$

$$\mathcal{L}_{\text{sparse}}^{\delta_{\text{var}}} = \frac{1}{|\mathcal{V}|(|\mathcal{V}| - 1)|A|} \sum_{i \neq j} \sum_{a_i} \text{Var}_{a_j} \left[ \delta_{ij}(\boldsymbol{\tau}_{ij}, \boldsymbol{a}_{ij}) \right]. \tag{34}$$

We scale these losses with a factor $\lambda_{\text{sparse}}$ and optimize them together with the TD loss. It is worth noting that these measurements and losses are not independent. For example, minimizing $\mathcal{L}_{\text{sparse}}^{\delta_{\text{var}}}$ would also reduce the variance of $q_{ij}$. Thus, in the next section, we consider all possible combinations between these measurements and losses.

**Observation-Based Approaches** In partial observable environments, agents sometimes need to coordinate with each other to share their observations and reduce their uncertainty about the true state (Wang et al., 2020). We can build our coordination graphs according to this intuition.

Agents use an attention model (Vaswani et al., 2017) to select the information they need. Specifically, we train fully connected networks $f_k$ and $f_q$ and estimate the importance of agent $j$'s observations to agent $i$ by:

$$\alpha_{ij} = f_k(\tau_i)^{\text{T}} f_q(\tau_j). \tag{35}$$

Then we calculate the global Q function as:

$$Q_{tot}(s, \boldsymbol{a}) = \frac{1}{|\mathcal{V}|} \sum_i q_i(\tau_i, a_i) + \sum_{i \neq j} \bar{\alpha}_{ij} q_{ij}(\boldsymbol{\tau}_{ij}, \boldsymbol{a}_{ij}), \tag{36}$$

where $\bar{\alpha}_{ij} = e^{\alpha_{ij}} / \sum_{i \neq j} e^{\alpha_{ij}}$. Then both $f_k$ and $f_q$ can be trained end-to-end with the standard TD loss. When building coordination graphs, given a sparseness controlling factor $\lambda$, we select $\lambda|\mathcal{V}|(|\mathcal{V}| - 1)$ edges with the largest $\bar{\alpha}_{ij}$ values.

### B.3 WHICH METHOD IS BETTER FOR LEARNING DYNAMICALLY SPARSE COORDINATION GRAPHS?

We show the learning curves of value-based implementations in Fig. 8-13 and compare our method ($\zeta_{ij}^{q_{\text{var}}}$ & $\mathcal{L}_{\text{sparse}}^{q_{\text{var}}}$) against the (attentional) observation-based method in Fig. 14. We can see that our proposed method generally performs better than the observation-based method, except for the task `Disperse`. Compared to other games, observations provided by `Disperse` can reveal all the game information. In this case, the observation-based method can make better use of local observations and can easily learn an accurate coordination graph.

## C THE SMAC BENCHMARK

On the SMAC benchmark, we compare our method against fully decomposed value function learning methods (VDN (Sunehag et al., 2018) & QMIX (Rashid et al., 2018)) and a deep coordination graph

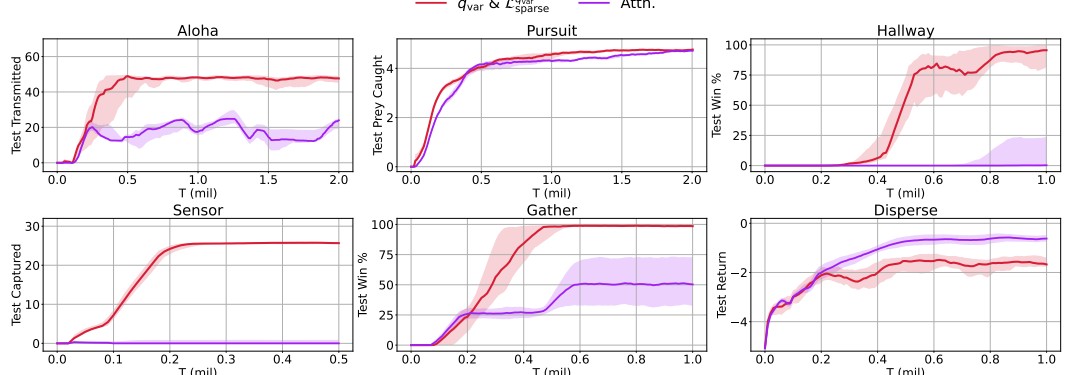

Figure 14: Performance comparison between our method ($\zeta_{ij}^{q_{\text{var}}}$ & $\mathcal{L}_{\text{sparse}}^{q_{\text{var}}}$) and the (attentional) observation-based approach on the MACO benchmark.

learning method (DCG (Böhmer et al., 2020)). Experiments are carried out on a hard map `5m_vs_6m` and a super hard map `MMM2`. For the baselines, we use the code provided by the authors and their default hyper-parameters settings that have been fine-tuned on the SMAC benchmark. We also notice that both our method and all the considered baselines are implemented based on the open-sourced codebase PyMARL[2], which further guarantees the fairness of the comparisons.

# D    ACTION REPRESENTATION LEARNING

As discussed in Sec. 3.3 of the main text, we use action representations to reduce the influence of utility difference function's estimation errors on graph structure learning. In this section, we describe the details of action representation learning (the related network structure is shown in Fig. 15). We use the technique proposed by Wang et al. (2021b) and learn an action encoder $f_e(\cdot; \theta_e)$: $\mathbb{R}^{|A|} \to \mathbb{R}^d$, parameterized by $\theta_e$, to map one-hot actions to a $d$-dimensional representation space. With the encoder, each action $a$ has a latent representation $z_a$, *i.e.*, $z_a = f_e(a; \theta_e)$. The representation $z_a$ is then used to predict the next observation $o_i'$ and the global reward $r$, given the current observation $o_i$ of an agent $i$, and the one-hot actions of other agents, $\boldsymbol{a}_{-i}$. This model is a forward model, which is trained by minimizing the following loss:

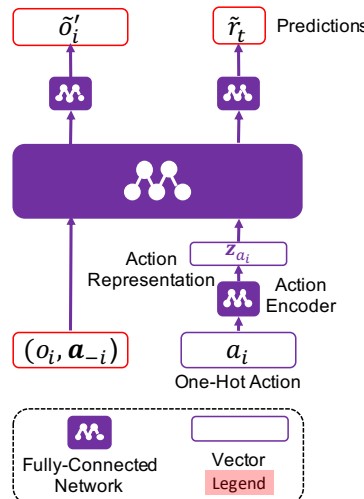

Figure 15: Framework for learning action representations, reproduced from Wang et al. (2021b).

$$
\begin{aligned}
\mathcal{L}_e(\theta_e, \xi_e) = \mathbb{E}_{(\boldsymbol{o}, \boldsymbol{a}, r, \boldsymbol{o}') \sim \mathcal{D}} \Big[ \sum_i \| p_o(z_{a_i}, o_i, \boldsymbol{a}_{-i}) - o_i' \|_2^2 \\
+ \lambda_e \sum_i \left( p_r(z_{a_i}, o_i, \boldsymbol{a}_{-i}) - r \right)^2 \Big],
\end{aligned}
$$
(37)

where $p_o$ and $p_r$ is the predictor for observations and rewards, respectively. We use $\xi_e$ to denote the parameters of $p_o$ and $p_r$. $\lambda_e$ is a scaling factor, $\mathcal{D}$ is a replay buffer, and the sum is carried out over all agents.

In the beginning, we collect samples and train the predictive model shown in Fig. 15 for $50K$ timesteps. Then policy learning begins and action representations are kept fixed during training. Since tasks in the MACO benchmark typically do not involve many actions, we do not use action representations when benchmarking our method. In contrast, StarCraft II micromanagement tasks usually have a large action space. For example, the map `MMM2` involves 16 actions, and a conventional deep Q-network requires 256 output heads for learning utility difference. Therefore, we equip our

---

[2]https://github.com/oxwhirl/pymarl

method with action representations to estimate the utility difference function when testing it on the SMAC benchmark.

# E    ARCHITECTURE, HYPERPARAMETERS, INFRASTRUCTURE, AND TIME COMPLEXITY

In CASEC, each agent has a neural network to estimate its local utility. The local utility network consists of three layers—a fully-connected layer, a 64 bit GRU, and another fully-connected layer—and outputs an estimated utility for each action. The utility difference function is also a 3-layer network, with the first two layers shared with the local utility function to process local action-observation history. The input to the third layer (a fully-connected layer) is the concatenation of the output of two agents' GRU layer. The local utilities and pairwise utility differences are summed to estimate the global action value (Eq. 11 in the paper).

For all experiments, the optimization is conducted using RMSprop with a learning rate of $5 \times 10^{-4}$, $\alpha$ of 0.99, RMSProp epsilon of 0.00001, and with no momentum or weight decay. For exploration, we use $\epsilon$-greedy with $\epsilon$ annealed linearly from 1.0 to 0.05 over $50K$ time steps and kept constant for the rest of the training. Batches of 32 episodes are sampled from the replay buffer. The default iteration number of the Max-Sum algorithm is set to 5. The communication threshold depends on the number of agents and the task, and we set it to 0.3 on the map `5m_vs_6m` and 0.35 on the map `MMM2`. We test the performance with different values ($1e{-}3$, $1e{-}4$, and $1e{-}5$) of the scaling weight of the sparseness loss $\mathcal{L}_{\text{sparse}}^{q_{\text{var}}}$ on `Pursuit`, and set it to $1e{-}4$ for both the MACO and SMAC benchmark. The whole framework is trained end-to-end on fully unrolled episodes. All experiments on StarCraft II use the default reward and observation settings of the SMAC benchmark.

All the experiments are carried out on NVIDIA Tesla P100 GPU. We show the estimated running time of our method on different tasks in Table 3 and 4. Typically, CASEC can finish 1M training steps within 8 hours on MACO tasks and in about 10 hours on SMAC tasks. In Table 5, we compare the computational complexity of action selection for CASEC and DCG, which is the bottleneck of both algorithms. CASEC is slightly faster than DCG by virtue of graph sparsity.

Table 3: Approximate running time of CASEC on tasks from the MACO benchmark.

| Aloha | Pursuit | Hallway | Sensor | Gather | Disperse |
|---|---|---|---|---|---|
| 13h (2M) | 17h (2M) | 7h (1M) | 4.5h (0.5M) | 6.5h (1M) | 8h (1M) |

Table 4: Approximate running time of CASEC on tasks from the SMAC benchmark.

| 5m_vs_6m | MMM2 |
|---|---|
| 18h (2M) | 21h (2M) |

Table 5: Average time (millisecond) for 1000 action selection phases of CASEC/DCG. CASEC uses a graph with sparseness 0.2 while DCG uses the full graph. To ensure a fair comparison, both Max-Sum/Max-Plus algorithms pass messages for 8 iterations. The batch size is set to 10.

|  | 5 actions | 10 actions | 15 actions |
|---|---|---|---|
| 5 agents | 2.90/3.11 | 3.15/3.39 | 3.42/3.67 |
| 10 agents | 3.17/3.45 | 3.82/4.20 | 5.05/5.27 |
| 15 agents | 3.41/3.67 | 5.14/5.4 | 7.75/8.02 |

# F    INFLUENCE OF CYCLES ON MAX-SUM

One limitation of our method is that we cannot guarantee cycle-free coordination graphs. Running Max-Sum on loopy graphs (graph that contains loops) may lead to sub-optimal actions being selected. In this section, we investigate the influence of cycles on the optimality of Max-Sum algorithm.

Specifically, we first compare the results of Max-Sum against optimal joint actions on `Aloha` from the MACO benchmark. To this end, we sample 1000 `Aloha` configurations. We then run Max-Sum under different sparseness degrees (with no guarantee of cycle-free graphs) and compare with optimal joint actions. Since finding the optimal action is NP-hard, we use a brute-force method and enumerate all possible joint actions and choose the one with the largest $Q_{tot}$ value.

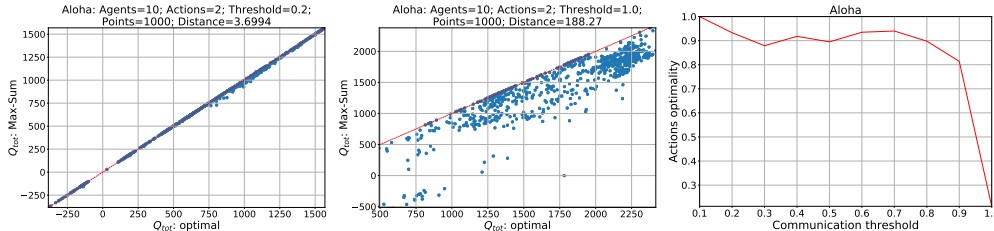

Figure 16: Compare actions selected by Max-Sum and the optimal joint action on 1000 different configurations of `Aloha`. **Left**: On sparse graphs with 20% edges. $Q_{tot}$ values of the actions are shown. **Middle**: On full graphs. $Q_{tot}$ values of the actions are shown. **Right**: How many actions selected by Max-Sum are optimal under different sparseness degrees.

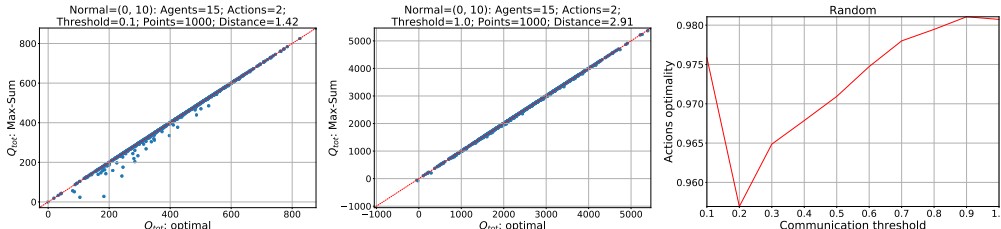

Figure 17: Compare actions selected by Max-Sum and the optimal joint action on 1000 random configurations. **Left**: On sparse graphs with 10% edges. $Q_{tot}$ values of the actions are shown. **Middle**: On full graphs. $Q_{tot}$ values of the actions are shown. **Right**: How many actions selected by Max-Sum are optimal under different sparseness degrees.

Results are shown in Fig. 16. We can see that Max-Sum on sparse graphs selects optimal actions in around 95% of the cases. $Q_{tot}$ values are also satisfactory, with most points falling near the line $y = x$. In comparison, the quality of Max-Sum solutions decreases significantly on full graphs, as shown in Fig. 16 middle and right.

We further investigate the case of random graphs. 1000 graphs are generated randomly with utility and payoff values conforming to a Gaussian distribution with a mean of 0 and a variance of 10, and we carry out experiments similar to those on `Aloha`. As shown in Fig. 17, we find that Max-Sum on both sparse and full graphs can select more than 95% optimal actions. The optimization objective, $Q_{tot}$, is also very close to the optimal value.

We can conclude that, although it can not guarantee optimality consistently, Max-Sum on sparse graphs can select the optimal action with a large probability on the tested cases. In comparison, optimal actions are less likely to be selected on full graphs. This may shed light on why CASEC can outperform DCG on some tasks. For future work, we plan to investigate how to learn cycle-free sparse coordination graphs so that action optimality can be guaranteed.

## G EMPIRICAL EVALUATION OF THE BOUND IN EQ. 5

To figure how loose the bound in Eq. 5 is, we randomly generate 10000 graphs, select the edge between agent 0 and 1, and put them into 100 bins according to the value of $\zeta_{01}^{q_{var}}$. In each bin, we calculate the number of graphs where the actions of agent 0 and 1 selected by Max-Sum keep unchanged after removing the edge between them. Also for each bin, we average the bound in Equation 5 of each graph instance. Then, in Fig. 18, we compare our bound against the frequency of unchanged actions. We observe that, on average, the bound is 36.9% lower than the real frequency.

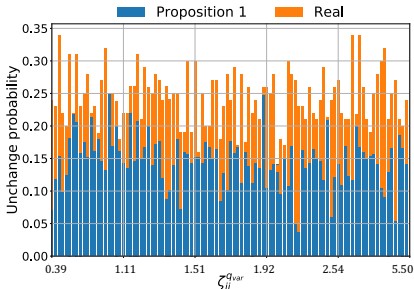

Figure 18: Comparison between the bound in Proposition 1 and the probability in real cases.

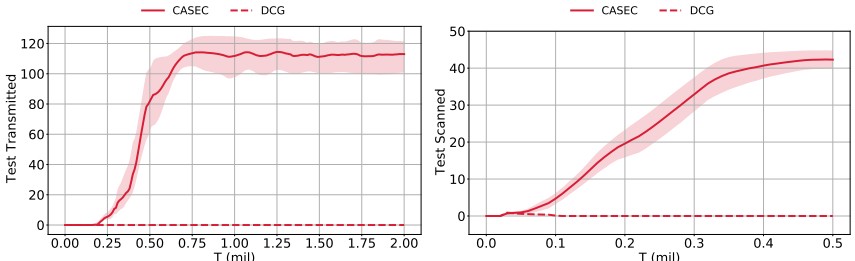

Figure 19: Comparison between CASEC and DCG on `Aloha` (**Left**) and `Sensor` (**Right**) with two times number of agents.

## H    TASKS WITH A LARGER NUMBER OF AGENTS

Intuitively, sparse graphs are expected to perform better in tasks with more agents. In this section, we compare CASEC with DCG on a large version of `Aloha` and `Sensor`.

The new version of `Aloha` has 20 agents in a $2 \times 10$ array. We compare CASEC against DCG in Fig. 19 left. We can see that DCG can no longer send any messages, but CASEC can send about 110 of them. For `Sensor`, there are 30 sensors and 6 targets. Results are shown in Fig. 19 right. We can see that DCG does not learn to scan any targets, while CASEC can capture about 40 of them. The gap between sparse and full coordination graphs is more significant on these tasks.

## I    DECIDE SPARSENESS ADAPTIVELY

A limitation of our method is that we fix the communication threshold when training. In this section, we study how to select the threshold adaptively and investigate the following two methods.

The first method is based on the observation that the performance of sparse graphs would degrade dramatically when the sparseness degree is below a certain value. To find this value, during testing, we check the performance of graphs with different sparseness degrees and select the degree below which the performance would drop. We change the threshold every 50K, 150K, and 200K training timesteps and show the performance in Fig. 20 (the first row). We can see that training with such an adaptive threshold performs similarly with the original CASEC algorithm after convergence and learns slightly better during the initial learning stage. The found threshold is smaller than the one that we get through a grid search.

The second method is based on Proposition 1. The intuition is that we can cut off the edges which exert limited influence on Max-Sum. Specifically, during testing, we count the number of edges that lead to different Max-Sum results with a probability smaller than 0.36 after being removed. The percentage of these edges is set as the communication threshold. Again, we change the threshold every 50K, 150K, and 200K training timesteps. The results are shown in Fig. 20 (the second row). This second method leads to higher final performance but learns slower initially. The adaptive threshold is less stable compared to the first method, and the selected thresholds are larger than the hand-crafted one.

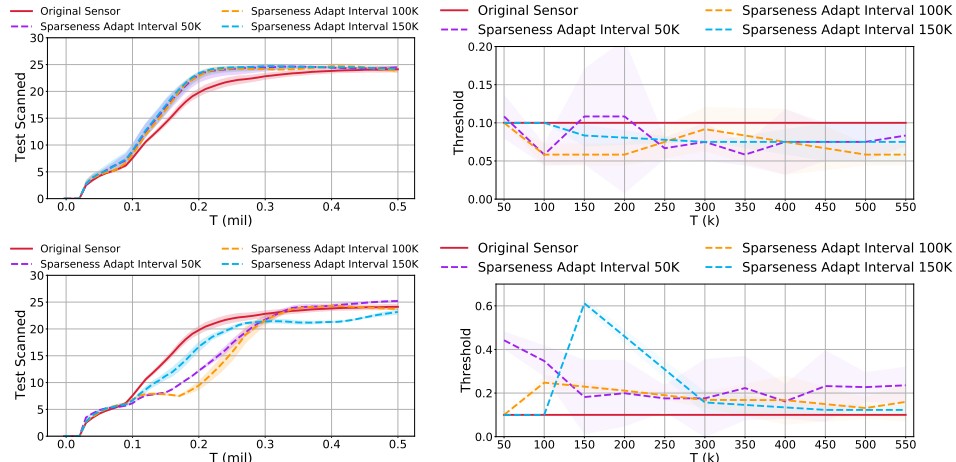

Figure 20: Learning curves (**Left**) and the changing process (**Right**) of the communication threshold of the two methods proposed in Appendix I.

For future work, it is an important question how to develop more principled methods that can find the minimum communication threshold which can guarantee learning performance.

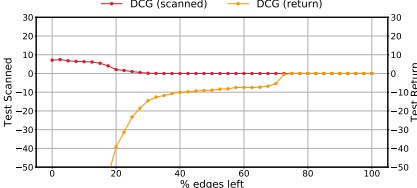

Figure 21: Performance of DCG on `Sensor` with different numbers of edges in the coordination graph.

