# OpenReview forum: "Context-Aware Sparse Deep Coordination Graphs"
_ICLR.cc/2022/Conference — ICLR 2022 Spotlight_

### Official Review · Reviewer_gGhv · 2021-11-01

**Correctness:** 3
**Technical Novelty And Significance:** 4
**Empirical Novelty And Significance:** 4
**Recommendation:** 8
**Confidence:** 4

**Main Review:**

**POSITIVE**

The paper is well written and the presented ideas are novel and interesting. The choice of scoring function for the edges is well motivated (although the reviewer was not able to follow the entire proof). In particular in the StarCraft II tasks the method shows a clear advantage over DCG.

**NEGATIVE**

(1) The analysis of Sensor is nice, but the benchmark seems to use the wrong evaluation metric. As the reward is linear in the number of agents that scan a target (as long as they are more than 2), the task does not specify whether it is better to scan few targets with many agents or many targets with few agents. All evaluation graphs (except Fig.1, middle) focus on the number of scanned targets, though, which is not optimized for. The fact that CASEC scans more targets than DCG does not mean anything, the fact that it collects more reward does. Moreover the conclusion "the gap between return and the number of scanned targets is larger without the sparseness loss" is somewhat weird, as adding the loss increases reward, but decreases scanned targets. The authors must address this issue. The reviewer recommends to clip the reward at 3 for 2 or more agents scanning the same target to get rid of this ambiguity and ensure all algorithms optimize the right metric.

(2) The authors claim "Intuitively, agent $i$ needs to coordinate its action selection with agent $j$ if agent $j$'s action exerts significant influence on the expected utility of agent $i$." However, an edge may also be important if agent $i$ needs access to the joint histories $\tau_{ij}$ to properly evaluate an action, similar to the "non-decentralizable task" in the DCG paper. Ideally the authors would disentangle this dependency in an additional experiment, but the possibility should at least be mentioned.

(3) The flat performance of DCG in the middle plot of Figure 1 is curious, as DCG seems to outperform CASEC (in terms of reward) for very few edges. Does this hold for no edges left as well? The authors comment this with "only the individual utility function contributes to action selection". Does this indicate that all payoff functions learned by DCG barely contribute, and wouldn't that mean they have almost zero variance? Shouldn't CASEC learn this kind of behavior first, as it is explicitly regularized towards it?

(4) There is no incentive for CASEC to use the utility functions, as only the payoffs' variances are punished. The network could therefore absorb the utilities into the payoffs. The authors could add a second regularization term that prevents this, e.g., a norm loss on the payoffs outputs with a very small regularization constant.

**DETAILS**

- The space of histories must be $T \equiv (\Omega \times A)^* \times \Omega$
- "and $q$ represents utility or payoff functions.": define clearer that $q$ is the (weighted) sum of all utility and payoff functions that condition on actions $a_g$.
- Mention where to find the proof of Proposition 1 in the main text.
- Why not use double-Q-learning in (eq.7)?
- Use $|\mathcal V|$ instead of $n$ in (eq.8) for consistency.
- You should explicitly mention that you train a fully connected graph, but perform action selection on the sparse graph.
- "and 2) accelerating the training of payoff function between target network updates to reduce the estimation errors." is unclear. This is about action representation. Say so.
- In the Sensor description of p.5: $k \geq 2$
- Mention in 6.2 whether you use the sparsity constants from 6.1
- "DCG outperforms CASEC on Hallway" is not significant
- CASEC reduces the representational capacity of DCG, which can explain why it learns faster, but you cannot argue that the CASEC solution is "beyond the representational capacity of the network" (p.8) of DCG.
- What is a "utility difference function" (p.9)?
- Mention the potential convergence issues earlier, e.g. p.3.

** POST-REBUTTAL **
The author's rebuttal incorporated this reviewers suggestion and the results appear much cleaner thanks to this. The discussion has convinced this reviewer that the paper should be accepted. The score has therefore be raised to 8.

**Summary Of The Paper:**

The paper introduces a technique called CASEC to train deep coordination graphs with sparse connectivity and a dynamic method to remove a fixed number of edges during inference. The latter removes edges in order of smallest variance of the corresponding payoff functions, which the authors motivate with a theoretical statement on the probability of changing the optimal action after removing an edge. CASEC is evaluated on a novel benchmark suit (MACO) for coordination dilemmas and StarCraft II micromanagement tasks. The proposed methods outperforms the baseline (DCG) in 2 out of 6 MACO tasks and shows in particular a clear advantage of the used action encoding method in StarCraft II.

**Summary Of The Review:**

The introduced method is novel and relevant. Although the reviewer was not able to follow the entire proof, the edge selection seems well motivated. Experiments are sometimes weirdly evaluated, but clear enough to say that in some cases using sparse graphs is an advantage. The reviewer would have liked to see more intuition when this is the case though.

In summary, this is a good paper that merits publication. The reviewer would consider increasing the score if the authors fix point (1) or convincingly explain why number of scanned targets is the correct evaluation metric.

---

> ### Author Response · Authors · 2021-11-19
> **Thanks for your valuable comments. We fix point (1) by presenting results on the suggested setting of Sensor. We also address other concerns.**
>
> - #### **Evaluation metrics of Sensor.**
>
> We change the game setting of $\mathtt{Sensor}$ as suggested by the reviewer. A reward of 3 is given when there are more than 2 agents scanning a target. Each scan still incurs a cost of -1. We update the case study section in the revised version, including the presentation and analyses of results.
>
> As for the two relevant concerns of the reviewer, we summarize the observation under this new setting as follows. (1) CASEC outperforms DCG in terms of both the return and the number of scanned targets. (2) Adding the sparseness loss increases the rewards of CASEC but scans a similar number of targets when the graph is sparse. This is because adding the sparseness loss removes useless scan actions.
>
> - #### **Disentangle the dependency of local observations.**
>
> In an additional experiment (observation-based methods, Appendix B.2, page 18), we disentangle the observation dependency. We assign weights to edges in coordination graphs via an attention mechanism conditioned on pairwise observations. Agents learn to attend to observations of other agents. We investigate the performance of this method on the MACO benchmark (Table 3, 4, 5 on page 16, 17,  20) and conclude that $\mathtt{Disperse}$ is characterized by _observation dependency_, while the performance gain achieved by CASEC on the other five tasks highlights the _action-value dependency_ among agents.
>
>
> - #### **The flat performance of DCG in the middle plot of Figure 1.**
>
> Under the new game setting, the return of CASEC is consistently larger than that of DCG, including the case where no edges are left. As for DCG, the number of scanned targets keeps unchanged when adding more edges, but return increases. This result indicates that payoff functions of DCG barely contribute to scanning targets, but contribute to removing useless scan actions.
>
> - #### **Details**
> - We update the history space to $T\equiv(\Omega\times A)^*\times \Omega$. (First paragraph, Sec. 2, page 2.)
>
> - We now define $q$ as the utility and payoff functions that condition on actions $a_g$ (page 3). In our paper, we consider pairwise edges, and $q$ is not the weighted sum of all utility and payoff functions that condition on actions $a_g$.
>
> - We add reference to the proof of Proposition 1. (Second paragraph, page 4.)
>
> - We update eq.8, using $|\mathcal{V}|$ instead of n. (Sec. 3.2, page 4.)
>
> - We now explicitly mention that we train a fully connected graph, but perform action selection on sparse graphs in the last paragraph, sec. 3.2, page 4.
>
> - We now explicitly state that we use action representation to "and 2) accelerating the training of payoff function between target network updates to reduce the estimation errors." (Third paragraph, sec. 3.3, page 4.)
>
> - The Sensor description on page 5 holds for both $k=2$ and $k>2$.
>
> - We mention that results in sec. 6.2 are obtained under the sparsity constants shown in Table 2. (Second paragraph, Sec. 6.1, page 8.)
>
> - We remove the statement that "DCG outperforms CASEC on Hallway." (The last paragraph, Sec. 6.2, page 8.)
>
> - We remove the claim about DCG ("which is beyond the representational capacity of the network." in the last paragraph, Sec. 6.2, page 8).
>
> - It should be "the utility and payoff function". We have updated it in the revised version.

---

> > ### Comment · Reviewer_gGhv · 2021-11-19
> > **Thanks for the changes**
> >
> > Thanks to the authors for changing their Sensor setup. The results look much cleaner now.
> >
> > However, now the left and the middle plot of Figure 1 seem to contradict each other. If the reviewer understand this correctly, in the left plot "DCG" corresponds to DCG with 100% edges in the middle plot. But the final performance (at 500k steps) of "DCG" on the left is (on average, the dashed line) 0 targets scanned and a return of 0, whereas DCG (100%) in the middle manages to scan 7.5 targets for around 7.5 return (although this is currently hard to read, see below). Please explain this difference.
> >
> > The two scales in the plots (targets scanned and return) are also very confusing. The scanned targets should be a hard upper bound on the return now, as scanning a target can at most achieve +1 reward and there are no other positive rewards. So it would make much more sense to scale the "targets scanned" axis to correspond to the "return" axis (or just use one axis scale for simplicity), which would improve the readability of Figure 1 considerably. Please change this or argue why this would be a bad idea.
> >
> > On a purely esthetic note, the fonts in Figure 1 are all way too small to be readable on a printout.

---

> > > ### Author Response · Authors · 2021-11-19
> > > **Thanks for the feedback. We provide clarification about Fig. 1. We also scale the axis of this figure and enlarge its fonts.**
> > >
> > > > Now the left and the middle plot of Figure 1 seem to contradict each other.
> > >
> > > Actually, they do not contradict. In the left plot, the dashed line shows the median performance instead of the average (we run 7 random seeds, and the boundaries of the shaded area are the 25\% and 75\% percentile). The middle plot of Fig. 1 shows the case of the best seed. Among 7 seeds, 5 of them scan 0 targets and get a return of 0, while the performance of the other two seeds is similar, which is shown in the middle plot. The median of these 7 seeds is 0 for both scanned targets and return.
> > >
> > > To eliminate confusion, we explain the meaning of lines and shaded areas in the latest version of our paper (second paragraph, Sec. 5, page 7). We also show the middle plot for other seeds of DCG (Fig. 21, page 24), where we can see that the number of scanned targets decreases and the return increases with more edges in the graph. DCG's payoff functions do so by removing all the scan actions induced by the individual utility functions.
> > >
> > > > The two scales in the plots (targets scanned and return) are also very confusing.
> > >
> > > We thank the reviewer for pointing out the problem of scale in Fig. 1 and the suggestion on how to improve it. We agree that it is a great idea to align the two scales. In the newest version, we update Fig. 1 according to the reviewer's suggestion, and we can see that the line of scanned targets is above the line of return now.
> > >
> > > > The fonts in Fig. 1 are all way too small to be readable on a printout.
> > >
> > > We enlarge the fonts in Figure 1 in the newest version.
> > >
> > > Thanks again for the reviewer's reply. We are happy to respond if the reviewer has any other questions.

---

> > > > ### Comment · Reviewer_gGhv · 2021-11-24
> > > > **Reviewer convinced**
> > > >
> > > > Thanks to the authors for explaining the difference between the two plots. The new results have convinced this reviewer, who will increase the score. Nonetheless, the caption of Figure 1 should include the information that the middle plot shows the best seeds from the left plot.

---

> > > > > ### Author Response · Authors · 2021-11-28
> > > > > **Thanks for the suggestions**
> > > > >
> > > > > We thank the reviewer for the feedback and suggestions, which help us significantly improve our paper.
> > > > >
> > > > > We will update the caption of Figure 1 in the next version to make it clear that we use the best seeds for the middle plot.

---

### Official Review · Reviewer_Y1rR · 2021-11-01

**Correctness:** 3
**Technical Novelty And Significance:** 3
**Empirical Novelty And Significance:** 3
**Recommendation:** 6
**Confidence:** 3

**Details Of Ethics Concerns:**

I reviewed this paper before, and multiple reviewers reported several very related works missing, i.e. Deep Implicit Coordination Graphs for Multi-agent Reinforcement Learning, AAMAS 2021). The authors completely ignored this criticism and did not incorporate it into a new version.

**Main Review:**

### Pros

- The problem is relevant;
- MACO benchmark is a valuable contribution to the community;
- Empirical results are good.
- Proposition 1 is an interesting addition to the paper (I did not check the proof).

### Cons

- The paper talks about "intensive and inefficient" message passing, however, it is unclear if a sparser graph leads to wall clock time reductions due to batching using GPUs.
- The paper hypothesises why CASEC outperforms DCG, however, it would merit a deeper analysis of the question. Personally, I think that this is the most interesting/important finding of the paper that sparsifying the graph leads to such performance improvements.
- The paper misses relevant work: Deep Implicit Coordination Graphs for Multi-agent Reinforcement Learning, AAMAS 2021 that use attention and graph networks to learn context-dependent coordination graphs doing an evaluation on SMAC as well.

### Questions/Comments

- Can you provide wallclock time comparison between DCG your method? It would be interesting to see how sparsifying the graph affects the real computation time.
- Can you provide the range of the fraction in Equation 5? I'd like to get a bit more intuition on about how loose the bound is.
- Is the result in Figure 2 consistent on different runs? Do all the seeds demonstrate this behaviour?
- The negative space in Figure 3 is just too large.
- "These results prove that our method can distinguish the most important edges". This statement is too strong, I don't think that empirical results can prove anything.
- The second paragraph on page 7 is an interesting analysis of DCG, this is a great find!
- Can you provide intuition on why DCG>CASEC on Pursuit?
- Can you include DCG in the TD error plot in Fig 5c?
- Why do you think there is a drop in performance for threshold = 0.5 in Figure 4 for Gather?

**Summary Of The Paper:**

The paper proposes to use the variance of the payoffs functions to learn dynamic coordination graphs, i.e. coordination graphs that might change in structure at each time step of the rollout. The empirical evaluation is carried out on a newly proposed MACO benchmark as well as some levels from the Starcraft II SMAC benchmark.

**Summary Of The Review:**

I reviewed this paper before, and the authors improved the manuscript and the paper reads much more nicely now. Empirical results are good, and the authors hypothesise on why this or that experiment went that way.

However, I give it a 5 (marginally below the threshold) due to the fact that the authors decided not to include the related work that the reviewers suggested, e.g. Deep Implicit Coordination Graphs for Multi-agent RL by Li et al. I am willing to increase the score if:
* authors provide a legitimate reason for not including relevant work into this submission, and
* include the relevant work and compare to it / explain why such comparison is not meaningful.

---

> ### Author Response · Authors · 2021-11-19
> **Thanks for your inspiring comments. We explain why we didn't compare to the relevant work, and discuss and compare to it in the new version. We also address other concerns.**
>
> ### About relevant works
>
> > **"Authors provide a legitimate reason for not including relevant work into this submission"**
>
> The main reason is that DICG [1] does not incorporate pairwise payoff functions. Graphs in DICG are used to mix observation of agents which serves as the input of fully-decomposed actors or centralized critics. Consequently, DICG still suffers from the relative generalization problem. In our ablation study, _Attn._ (described in Appendix B.2), we use the DICG-style attention mechanism to assign weights to payoff functions. We empirically compare against this ablation in Table 5 (page 20).
>
> As suggested by the reviewer, in the revised version of our paper, we discuss and compare against DICG.
>
> > **"Include the relevant work and compare to it."**
>
> We discuss the relevant work [1, 2] in Section 2 of the revised version (Page 3). Moreover, empirical comparison on both the MACO and SMAC benchmark is shown in Fig. 3 (Page 7) and 5 (Page 9). Since [1, 2] do not have pair-wise payoff functions, we observe that it performs similarly to other fully decomposed methods like QMIX and weighted-QMIX.
>
> ### Questions
>
> > **"Wall clock time comparison between DCG and your method."**
>
> In Table 9 (page 22), we compare the wall clock time for each action selection under the different numbers of agents (5, 10, 15) and the different number of actions (5, 10, 15). We observe that CASEC consistently reduces wall clock time, typically by about 0.2ms for each action selection, regardless of the number of agents and actions. Batching using GPUs does have effects: when CASEC cuts off 80\% edges, it saves 9.05\% wall clock time (10 agents, 10 actions).
>
> > **"How loose the bound in Equation 5 is?"**
>
> We empirically compare this bound with probability in real cases. We randomly generate 10000 graphs, select the edge between agent 0 and 1, and put them into 100 bins according to the value of $\zeta_{01}^{q_{var}}$. In each bin, we calculate the number of instances where the actions of agent 0 and 1 selected by Max-Sum keep unchanged after removing the edge between them. Also for each bin, we average the bound in Equation 5 of each instance. Then, in Fig. 19 (Appendix H, page 24), we compare our bound against the frequency of unchanged actions. We observe that, on average, our lower bound is 36.9\% lower than the real frequency.
>
> > **"Is the result in Figure 2 consistent on different runs? Do all the seeds demonstrate this behaviour?"**
>
> Yes, we run 5 seeds and observe consistent behavior.
>
> > **"The negative space in Figure 3 is just too large."**
>
> We now narrow this space.
>
> > **"These results prove that our method can distinguish the most important edges". This statement is too strong, I don't think that empirical results can prove anything.**
>
> We update this statement. In the revised version, we use the following claim: These results demonstrate that our method can distinguish the most important edges on $\mathtt{Sensor}$.
>
> > **"Can you provide intuition on why DCG > CASEC on Pursuit?"**
>
> We think this is because of some implementation level differences or hyper-parameter settings. In Fig. 9 (page 17), we see that CASEC with the sparseness value of 1.0 (i.e., the full graph) also underperforms DCG in the initial phase. This observation indicates that graph sparseness does not matter in this phase.
>
> Moreover, for other training phases, we find that CASEC outperforms DCG after convergence, especially when there are more agents. On $\mathtt{Pursuit}$ with 20 agents, CASEC agents catch 9.731 preys on average while DCG catches 9.348 (5 random seeds).
>
> > **"Can you include DCG in the TD error plot in Fig 5c?"**
>
> In the updated version, we include DCG (with rank 1 approximation) in the TD error plot. As expected, the TD error of DCG is consistently larger than that of _Full Graph with Action Representations_ during the training process.
>
> [1] Li, S., Gupta, J.K., Morales, P., Allen, R. and Kochenderfer, M.J., 2021, May. Deep Implicit Coordination Graphs for Multi-agent Reinforcement Learning. In Proceedings of the 20th International Conference on Autonomous Agents and MultiAgent Systems (pp. 764-772).
>
> [2] Navid Naderializadeh, Fan H. Hung, Sean Soleyman, Deepak Khosla. "Graph Convolutional Value Decomposition in Multi-Agent Reinforcement Learning". arXiv:2010.04740

---

> > ### Comment · Reviewer_Y1rR · 2021-11-19
> > **response**
> >
> > I acknowledge the authors' response and increasing the score after the rebuttal.
> >
> > >>> This observation indicates that graph sparseness does not matter in this phase.
> >
> > I believe this is a central question and further investigations of it would lead to a much stronger paper.

---

> > > ### Author Response · Authors · 2021-11-23
> > > **Thanks for the feedback**
> > >
> > > We thank the reviewer for the suggestions and the feedback.
> > >
> > > This question is a very interesting future work and we will continue studying it.

---

### Official Review · Reviewer_7Rh6 · 2021-11-05

**Correctness:** 3
**Technical Novelty And Significance:** 3
**Empirical Novelty And Significance:** 3
**Recommendation:** 6
**Confidence:** 3

**Main Review:**

Overall speaking, I found the paper interesting to read and easy to follow. In comparison with existing literature on in cooperative multi-agent learning, I'm convinced that, the approach proposed this paper indeed adds valuable extensions in multiple dimensions and it would be quite beneficial to the research community in this area. As far as I know, the technique proposed in this paper is novel, and the demonstrated results of this proposed method/technique seem to be encouraging and promising. The presented MACO benchmark in this paper can also be valuable addition to the research community in this area, and the empirical evaluation of the proposed method v.s. existing SOTA methods on MACO benchmark and StarCraft II micromanagement benchmark, also provides some reasonable justification and illustration of the superior performance of the proposed new method.

The two major limitations of the proposed approach (also already pointed out by the authors in the conclusions section), especially about the first one about no guarantee of cycle-free graph and thus might select sub-optional actions, is a bit concerning, and it would be valuable if the authors could provide some more comprehensive study on this limitation and evaluate more accurately how severe this limitation is (based on the current write-up, we are not fully clear how severe this limitation would be for general scenarios) . We want to avoid the case that the good numbers and examples in the experimental section are specially hand-crafted or carefully selected in a way to minimize the impact of this limitation of no cycle-free guarantee and to favor the proposed methods to show off its advantages. If that study wold takes too much extra time and efforts, I won't object for acceptance of the paper based on the current available results, since I think these results already could be considered as significant contributions to this research area, and I'm ok for the authors to leave those study as future research work. But a comprehensive study about this limitation and some more justification/explanation about it would definitely makes the paper stronger.




**Summary Of The Paper:**

In this paper, the authors study how to learn dynamic sparse coordination graphs, which is a long-standing problem in cooperative MARL. In particular, they propose a novel deep method that learns context-aware sparse coordination graphs adaptive to the dynamic coordination requirements, and evaluate the proposed method against existing methods on MACO benchmark, as well as StarCraft II micromanagement benchmark.

**Summary Of The Review:**

The paper is well-written and seems to be of significant contribution to this research area. There are some limitations that are a bit concerning and it would be significantly better if the authors could provide more comprehensive study and explanations on those main limitations, but if not possible due to time limitation, I won't object for acceptance of the paper based on the current results and leave those potential improvements as future research work.

---

> ### Author Response · Authors · 2021-11-19
> **Thanks for your valuable comments. We carry out a comprehensive study to evaluate the influence of cycles in sparse coordination graphs.**
>
> > **"No guarantee of the cycle-free graph and thus might select sub-optimal actions."**
>
> Selecting optimal actions in coordination graphs with cycles is a sub-problem in the field of loopy belief propagation. Since the theoretical analysis of optimality in loopy cases is largely absent in this field, previous work studies this problem mainly from an empirical perspective [1]. Along with this line of research, we empirically study the influence of cycles on the optimality of the Max-Sum algorithm. The details can be found in Appendix G of the revised version.
>
> Specifically, we first compare the results of Max-Sum against optimal joint actions on $\mathtt{Aloha}$ from the MACO benchmark. To this end, we sample 1000 $\mathtt{Aloha}$ configurations. We then run Max-Sum under different sparseness degrees (with no guarantee of cycle-free graphs) and compare with optimal joint actions. Since finding the optimal action is NP-hard in this case, we use a brute-force method and enumerate all possible joint actions and choose the one with the largest $Q_{tot}$ value.
>
> Two metrics are used to evaluate the optimality of Max-Sum: (1) The difference in terms of the optimization object, $Q_{tot}$; (2) How many actions selected by Max-Sum are optimal.
>
> Results are shown in Fig. 17 (page 23). We can see that Max-Sum on sparse graphs selects optimal actions in around 95\% of the cases. $Q_{tot}$ values are also satisfactory, with an average error of 3.70. In comparison, the quality of Max-Sum solutions decreases significantly on full graphs, with an average error of 188.47 and only 21.2\% actions being optimal (Fig. 17 middle and right).
>
> We further investigate the case of random graphs. 1000 graphs are generated randomly with utility and payoff values conforming to a Gaussian distribution with the mean of 0 and the variance of 10, and we carry out experiments similar to those on $\mathtt{Aloha}$. As shown in Fig. 18 (page 23), we find that Max-Sum on both sparse and full graphs can select more than 95\% optimal actions. The optimization objective, $Q_{tot}$, is also very close to the optimal value.
>
> We can see that, although it can not guarantee optimality consistently, Max-Sum on sparse graphs can select the optimal action with a large probability on the tested cases. In comparison, optimal actions are less likely to be selected on full graphs. This observation is in line with previous work in the field (e.g., [2], especially exercise 4.7). We thus conclude that the influence of cycles on Max-Sum is limited on sparse graphs. Building cycle-free coordination graphs can eliminate such influence and is a promising direction for future research. Detailed experimental setup, results, and analysis can be found in Appendix G.
>
> [1] Murphy, K., Weiss, Y. and Jordan, M.I., 2013. Loopy belief propagation for approximate inference: An empirical study. arXiv preprint arXiv:1301.6725.
>
> [2] Judea Pearl. Probabilistic Reasoning in Intelligent Systems: Networks of Plausible Inference. Morgan Kaufmann, 1988.

---

> ### Author Response · Authors · 2021-11-23
> **A study on the adaptive selection of the communication threshold. Two methods are proposed and investigated. We achieve similar learning performance and smaller thresholds than the original CASEC algorithm.**
>
> The second limitation that the reviewer is concerned about is that we fix the communication threshold when training. In Appendix J of the revised version, we study how to select the threshold adaptively and investigate the following two methods.
>
> The first method is based on the observation that the performance of sparse graphs would degrade dramatically when the sparseness degree is below a certain value. To find this value, during testing, we check the performance of graphs with different sparseness degrees and select the degree below which the performance would drop. We change the threshold every $50$K, $150$K, and $200$K training timesteps and show the performance on $\mathtt{Sensor}$ in Fig. 22 (the first row) (page 25). We can see that training with such an adaptive threshold performs similarly with the original CASEC algorithm after convergence and learns slightly better during the initial learning stage. The found threshold is smaller than the one that we get through a grid search.
>
> The second method is based on Proposition 1. The intuition is that we can cut off the edges which exert limited influence on Max-Sum. Specifically, during testing, we count the number of edges that lead to different Max-Sum results with a probability smaller than 0.36 after being removed. The percentage of these edges is set as the communication threshold. Again, we change the threshold every $50$K, $150$K, and $200$K training timesteps. The results on $\mathtt{Sensor}$ are shown in Fig. 22 (the second row) (page 25). This second method leads to higher final performance but learns slower initially. The adaptive threshold is less stable compared to the first method, and the selected thresholds are larger than the hand-crafted one.
>
> We hope that these results can inspire more studies in this direction. For future work, it is an important question how to develop more principled methods that can find the minimum communication threshold which can guarantee learning performance.

---

> > ### Author Response · Authors · 2021-11-28
> > **Summary of updates for addressing the two limitations.**
> >
> > Dear reviewer 7Rh6,
> >
> > Thank you for the valuable suggestions. To address your concerns, we make the following revisions to our submission:
> >
> > (1) In Appendix G, we comprehensively study whether the influence of cycles is severe. The main conclusion is that although Max-Sum on full graphs is not precise enough, on sparse graphs, over 90% of the actions are optimal for both MACO and randomly generated tasks;
> >
> > (2) In Appendix J, we propose two methods to adaptively control the sparseness of graphs. One of them achieves similar performance and smaller sparseness degrees to the grid-searched ones, and another method also obtains comparable performance.
> >
> > We are wondering whether these studies have addressed your concerns. If they did not, we are happy to carry out further investigation and improve our paper.

---

> ### Comment · Reviewer_7Rh6 · 2021-11-29
> **All my questions have been addressed**
>
> Thanks for the helpful clarifications. All my questions have been addressed.

---

### Official Review · Reviewer_pXtp · 2021-11-05

**Correctness:** 4
**Technical Novelty And Significance:** 4
**Empirical Novelty And Significance:** 3
**Recommendation:** 8
**Confidence:** 4

**Main Review:**


**Strengths**

Earlier works incorporating coordination graphs in the function approximation case which either are limited to static coordination graphs (like DCG) or were learning a "soft" version of fully connected coordination graph [1] (with graph neural network based action inference). In comparison, the current work provides a novel alternative for the problem of coordination graph learning. While the paper doesn't mention it, the payoff variance based edge selection method might be an interesting case for resolving some of the future work mentioned in [2]. In a sense the MCTS setting might be a better test because it would resolve the chicken and egg problem that comes from trying to learn the payoff functions and can avoid the need for action encoders with auxiliary losses to stabilize learning. The paper has an expansive set of experiments with fair number of agents.

**Weaknesses**

Works like [1] should be mentioned in related work. While there are some ablations, so it's not a big weakness, but it would be useful to know for example whether the action encoder trick helps say qmix or other value factorization methods. I would expect sparsity to help even more in larger number of agent environments and it would be great if MACO allows to quickly evaluate that.

[1] Navid Naderializadeh, Fan H. Hung, Sean Soleyman, Deepak Khosla. "Graph Convolutional Value Decomposition in Multi-Agent Reinforcement Learning". arXiv:2010.04740

[2] Shushman Choudhury, Jayesh K Gupta, Peter Morales, Mykel J Kochenderfer. "Scalable Anytime Planning for Multi-Agent MDPs". AAMAS 2021.

**Summary Of The Paper:**

The paper focuses on the problem of cooperative multi-agent Reinforcement learning and proposes a novel way to learn the dynamic (state dependent) coordination graph for the joint action selection from factorized joint value representations. The key idea here is to use the variance in payoff function estimates can be a good indicator for whether a coordination graph edge should be present or not. However, given the deep RL goal is learn these payoff functions there is a cyclic dependency between errors from payoff estimation and coordination graph estimation. The paper therefore also proposes an alternative action representation scheme that helps mitigate some of these problems. The paper evaluates the proposed approach on many classic coordination problems as well as subset of SMAC benchmark tasks.

**Summary Of The Review:**

Overall it's a very interesting paper and would recommend acceptance just for the new variance based metric for coordination edge selection.

---

> ### Author Response · Authors · 2021-11-19
> **Thanks for your valuable comments. We discuss and compare against related work and evaluate sparse graphs in an environment with a larger number of agents.**
>
> - **About related work.**
>
> Thanks for pointing out related work [1]. In the revised version, we discuss this paper in Sec. 2 and compare with it on both the MACO (Fig. 3) and SMAC (Fig. 5) benchmark. Instead of representing pair-wise payoff functions, graphs in [1] are used to mix fully decomposed local utility functions. Therefore, it is not surprising to observe that it performs similarly to other fully decomposed baselines like QMIX and weighted-QMIX.
>
> - **Tasks with a larger number of agents.**
>
> We compare CASEC with DCG on a large version of $\mathtt{Aloha}$ and $\mathtt{Sensor}$.
>
> The new version of $\mathtt{Aloha}$ has $20$ agents in a $2\times 10$ array. Performance of CASEC and DCG is shown in Fig. 20 left (page 24). We can see that DCG can no longer send any messages, but CASEC can send about $110$ of them. For $\mathtt{Sensor}$, there are $30$ sensors and $6$ targets. Results are shown in Fig. 20 right (page 24). We can see that DCG does not learn to scan any targets, while CASEC can capture about $40$ of them. As expected by the reviewer, the gap between sparse and full coordination graphs is more significant on tasks with more agents.
>
> [1] Navid Naderializadeh, Fan H. Hung, Sean Soleyman, Deepak Khosla. "Graph Convolutional Value Decomposition in Multi-Agent Reinforcement Learning". arXiv:2010.04740

---

### Decision · Program_Chairs · 2022-01-20

**Decision:**

Accept (Spotlight)

**Comment:**

All reviewers found that the paper offers interesting contributions for multi-agent RL and favour acceptance of the paper. The strengths of the paper are summarized below:
- Good algorithmic contribution
- Offers a new set of benchmark tasks for coordination in MARL settings
- Exhaustive experiments on complex tasks with a reasonable number of agents
- All the issues raised by the reviewers (missing references, missing discussion of limitations...) have been satisfactorliy addressed.

I therefore join the reviewers in the recommendation to accept the paper.